# On the Minimax Regret for Online Learning with Feedback Graphs

**Khaled Eldowa**\*
Università degli Studi di Milano, Milan, Italy
`khaled.eldowa@unimi.it`

**Emmanuel Esposito**\*
Università degli Studi di Milano, Milan, Italy
& Istituto Italiano di Tecnologia, Genoa, Italy
`emmanuel@emmanuelesposito.it`

**Tommaso Cesari**
University of Ottawa, Ottawa, Canada
`tcesari@uottawa.ca`

**Nicolò Cesa-Bianchi**
Università degli Studi di Milano, Milan, Italy
& Politecnico di Milano, Milan, Italy
`nicolo.cesa-bianchi@unimi.it`

## Abstract

In this work, we improve on the upper and lower bounds for the regret of online learning with strongly observable undirected feedback graphs. The best known upper bound for this problem is $\mathcal{O}\big(\sqrt{\alpha T \ln K}\big)$, where $K$ is the number of actions, $\alpha$ is the independence number of the graph, and $T$ is the time horizon. The $\sqrt{\ln K}$ factor is known to be necessary when $\alpha = 1$ (the experts case). On the other hand, when $\alpha = K$ (the bandits case), the minimax rate is known to be $\Theta\big(\sqrt{KT}\big)$, and a lower bound $\Omega\big(\sqrt{\alpha T}\big)$ is known to hold for any $\alpha$. Our improved upper bound $\mathcal{O}\big(\sqrt{\alpha T(1 + \ln(K/\alpha))}\big)$ holds for any $\alpha$ and matches the lower bounds for bandits and experts, while interpolating intermediate cases. To prove this result, we use FTRL with $q$-Tsallis entropy for a carefully chosen value of $q \in [1/2, 1)$ that varies with $\alpha$. The analysis of this algorithm requires a new bound on the variance term in the regret. We also show how to extend our techniques to time-varying graphs, without requiring prior knowledge of their independence numbers. Our upper bound is complemented by an improved $\Omega\big(\sqrt{\alpha T(\ln K)/(\ln \alpha)}\big)$ lower bound for all $\alpha > 1$, whose analysis relies on a novel reduction to multitask learning. This shows that a logarithmic factor is necessary as soon as $\alpha < K$.

## 1 Introduction

Feedback graphs [29] provide an elegant interpolation between two popular online learning models: multiarmed bandits and prediction with expert advice. When learning with an undirected feedback graph $G$ over $K$ actions, the online algorithm observes not only the loss of the action chosen in each round, but also the loss of the actions that are adjacent to it in the graph. Two important special cases of this setting are: prediction with expert advice (when $G$ is a clique) and $K$-armed bandits (when $G$ has no edges). When losses are generated adversarially, the regret in the feedback graph setting with strong observability has been shown to scale with the independence number $\alpha$ of $G$. Intuitively, denser graphs, which correspond to smaller independence numbers, provide more feedback to the learner, thus enabling a better control on regret. More specifically, the best known upper and lower bounds on the regret after $T$ rounds are $\mathcal{O}\big(\sqrt{\alpha T \log K}\big)$ and $\Omega\big(\sqrt{\alpha T}\big)$ [3, 4]. It has been known for three decades that this upper bound is tight for $\alpha = 1$ (the experts case, [9, 10]). When $\alpha = K$

---

\*Equal contribution.

37th Conference on Neural Information Processing Systems (NeurIPS 2023).

(the bandits case), the lower bound $\Omega(\sqrt{KT})$—which has also been known for nearly three decades [7, 8]—was matched by a corresponding upper bound $\mathcal{O}(\sqrt{KT})$ only in 2009 [5]. These results show that in feedback graphs, the logarithmic factor $\sqrt{\log K}$ is necessary (at least) for the $\alpha = 1$ case, while it must vanish from the minimax regret as $\alpha$ grows from 1 to $K$, but the current bounds fail to capture this fact. In this work, we prove new upper and lower regret bounds that for the first time account for this vanishing logarithmic factor.

To prove our new upper bound, we use the standard FTRL algorithm run with the $q$-Tsallis entropy regularizer ($q$-FTRL for short). It is well-known [1] that for $q = \frac{1}{2}$ this algorithm (run with appropriate loss estimates) achieves regret $\mathcal{O}(\sqrt{KT})$ when $\alpha = K$ (bandits case), while for $q \to 1^-$ the same algorithm (without loss estimates) recovers the bound $\mathcal{O}(\sqrt{T \log K})$ when $\alpha = 1$ (experts case). When $G$ contains all self-loops, we show in Theorem 1 that, if $q$ is chosen as a certain function $q(\alpha, K)$, then $q(\alpha, K)$-FTRL, run with standard importance-weighted loss estimates, achieves regret $\mathcal{O}(\sqrt{\alpha T(1 + \log(K/\alpha))})$. This is a strict improvement over the previous bound, and matches the lower bounds for bandits and experts while interpolating the intermediate cases. This interpolation is reflected by our choice of $q$, which goes from $\frac{1}{2}$ to 1 as $\alpha$ ranges from 1 to $K$. The main technical hurdle in proving this result is an extension to arbitrary values of $q \in \left[\frac{1}{2}, 1\right)$ of a standard result— see, e.g., [29, Lemma 3]—that bounds in terms of $\alpha$ the variance term in the regret of $q$-FTRL. In Theorem 2, using a modified loss estimate, this result is extended to any strongly observable undirected graph [2], a class of feedback graphs in which some of the actions do not reveal their loss when played. In Theorem 3, we show via a doubling trick that our new upper bound can also be obtained (up to constant factors) without the need of knowing (or computing) $\alpha$. As the resulting algorithm is oblivious to $\alpha$, our analysis also applies to arbitrary sequences of graphs $G_t$, where $K$ is constant but the independence number $\alpha_t$ of $G_t$ can change over time, and the algorithm observes $G_t$ only after choosing an action (the so-called uninformed case). In this setting, the analysis of the doubling trick is complicated by the non-trivial dependence of the regret on the sequence of $\alpha_t$.

We also improve on the $\Omega(\sqrt{\alpha T})$ lower bound by proving a new $\Omega(\sqrt{\alpha T \log_\alpha K})$ lower bound for all $\alpha > 1$. This is the first result showing the necessity—outside the experts case—of a logarithmic factor in the minimax regret for all $\alpha < K$. Our proof uses a stochastic adversary generating both losses and feedback graphs via i.i.d. draws from a joint distribution. This sequence of losses and feedback graphs can be used to define a hard instance of the multi-task bandits problem, a variant of the combinatorial bandits framework [12]. We then prove our result by adapting known lower bounding techniques for multi-task bandits [6]. Note that for values of $\alpha$ bounded away from 2 and $K$, the logarithmic factor $\log_\alpha K$ in the lower bound is smaller than the corresponding factor $1 + \log(K/\alpha)$ in the upper bound. Closing this gap remains an open problem.

## 1.1 Additional related work

Several previous works have used the $q$-Tsallis regularizer with $q$ tuned to specific values other than $\frac{1}{2}$ and 1. For example, in [35, Section 4], $q$ is chosen as a function of $K$ to prove a regret bound of $\mathcal{O}(\sqrt{\alpha T(\log K)^3})$ for any strongly observable directed feedback graph, which shaves off a $\log T$ factor compared to previous works. This bound is worse than the corresponding bounds for undirected graphs because the directed setting is harder. Specific choices of $q$ have been considered to improve the regret in settings of online learning with standard bandit feedback. For example, the choice $q = \frac{2}{3}$ was used in [31] to improve the analysis of regret in bandits with decoupled exploration and exploitation. Regret bounds for arbitrary choices of $q$ are derived in [36, 23] for a best-of-both-worlds analysis of bandits, though $q = \frac{1}{2}$ remains the optimal choice. The $\frac{1}{2}$-Tsallis entropy and the Shannon entropy ($q = 1$) regularizers have been combined before in different ways to obtain best-of-both-worlds guarantees for the graph feedback problem [18, 22]. The idea of using values of $q \in (\frac{1}{2}, 1)$ for feedback graphs is quite natural and has been brought up before, e.g., in [32], but achieving an improved dependence on the graph structure by picking a suitable value of $q$ has not been, to the best of our knowledge, successfully pursued before. On the other hand, an approach based on a similar use of the $q$-Tsallis regularizer has been employed by [26] for the problem of multiarmed bandits with sparse losses to achieve a $\mathcal{O}(\sqrt{sT \ln(K/s)})$ regret bound, where $s$ is the maximum number of nonzero losses at any round.

Our lower bound is reminiscent of the $\Omega(\sqrt{KT \log_K N})$ lower bound proved in [33] for the problem of bandits with expert advice (with $N \geq K$ being the number of experts); see also [17] and [34]. In

that problem, at each time step, experts suggest distributions over actions to the learner, whose regret is computed against the best expert in hindsight. Although the two settings are different, the variant of the multitask bandit problem that our lower bound construction simulates is the same as the one used in the proof of [17, Theorem 7].

## 2   Problem Setting

For any integer $n \geq 1$, let $[n] = \{1, \ldots, n\}$. We consider the following game played over $T$ rounds between a learner with action set $V = [K]$ and the environment. At the beginning of the game, the environment secretly selects a sequence of losses $(\ell_t)_{t \in [T]}$, where $\ell_t \colon V \to [0, 1]$,[2] and a sequence of undirected graphs $(G_t)_{t \in [T]}$ over the set of actions $V$, that is, $G_t = (V, E_t)$. At any time $t$, the learner selects an arm $I_t$ (possibly at random), then pays loss $\ell_t(I_t)$ and observes the feedback graph $G_t$ and all losses $\ell_t(i)$ of neighbouring actions $i \in N_{G_t}(I_t)$, where $N_{G_t}(i) = \{j \in V \,:\, (i, j) \in E_t\}$ (see Online Protocol 1). In this work, we only focus on strongly observable graphs [2]. An undirected graph $G$ is strongly observable if for every $i \in V$, at least one of the following holds: $i \in N_G(i)$ or $i \in N_G(j)$ for all $j \neq i$.

The performance of the learner is measured by the regret

$$R_T = \mathbb{E}\left[\sum_{t=1}^{T} \ell_t(I_t)\right] - \min_{i \in [K]} \sum_{t=1}^{T} \ell_t(i) \ .$$

where the expectation is over the learner's internal randomization.

---

**Online Protocol 1** Online learning with feedback graphs

---

   **environment:** (hidden) losses $\ell_t \colon V \to [0, 1]$ and graphs $G_t = (V, E_t)$, for all $t = 1, \ldots, T$
   **for** $t = 1, \ldots, T$ **do**
      The learner picks an action $I_t \in V$ (possibly at random)
      The learner incurs loss $\ell_t(I_t)$
      The learner observes losses $\left\{\big(i, \ell_t(i)\big) : i \in N_{G_t}(I_t)\right\}$ and graph $G_t$
   **end for**

---

We denote by $\Delta_K$ the simplex $\left\{p \in [0, 1]^K \,:\, \|p\|_1 = 1\right\}$. For any graph $G$, we define its independence number as the cardinality of the largest set of nodes such that no two nodes are neighbors, and denote it by $\alpha(G)$. For simplicity, we use $N_t$ to denote the neighbourhood $N_{G_t}$ in the graph $G_t$ and we use $\alpha_t$ to denote the independence number $\alpha(G_t)$ of $G_t$ at time $t$.

## 3   FTRL with Tsallis Entropy for Undirected Feedback Graphs

As a building block, in this section, we focus on the case when all the feedback graphs $G_1, \ldots, G_T$ have the same independence number $\alpha_1 = \cdots = \alpha_T = \alpha$, whereas the general case is treated in the next section. For simplicity, we start with the assumption that all nodes have self-loops: $(i, i) \in E_t$ for all $i \in V$ and all $t$. We later lift this requirement and show that the regret guarantees that we provide can be extended to general strongly observable undirected feedback graphs, only at the cost of a constant multiplicative factor.

The algorithm we analyze is $q$-FTRL (described in Algorithm 1), which is an instance of the follow the regularized leader (FTRL) framework—see, e.g., [30, Chapter 7]—with the (negative) $q$-Tsallis entropy

$$\psi_q(x) = \frac{1}{1-q}\left(1 - \sum_{i \in V} x(i)^q\right) \qquad \forall x \in \Delta_K \ ,$$

as the regularizer, whose parameter $q \in (0, 1)$ can be tuned according to our needs. Since we do not observe all the losses in a given round, the algorithm makes use of unbiased estimates for the losses. When all self-loops are present, we define the estimated losses in the following standard manner. Let

---

[2]For notational convenience, we will sometimes treat the loss functions $\ell_t \colon V \to [0, 1]$ as vectors with components $\ell_t(1), \ldots, \ell_t(K)$.

$I_t$ be the action picked at round $t$, which is drawn from the distribution $p_t \in \Delta_K$ maintained by the algorithm, the loss estimate for an action $i \in V$ at round $t$ is given by

$$\widehat{\ell}_t(i) = \frac{\ell_t(i)}{P_t(i)} \mathbb{I}\{I_t \in N_t(i)\} \ , \tag{1}$$

where $P_t(i) = \mathbb{P}(I_t \in N_t(i)) = \sum_{j \in N_t(i)} p_t(j)$. This estimate is unbiased in the sense that $\mathbb{E}_t[\widehat{\ell}_t(i)] = \ell_t(i)$ for all $t \in [T]$ and all $i \in V$, where we denote $\mathbb{E}_t[\cdot] = \mathbb{E}[\cdot \mid I_1, \ldots, I_{t-1}]$.

---

**Algorithm 1** $q$-FTRL for undirected feedback graphs

---

    **input:** $q \in (0,1), \eta > 0$
    **initialization:** $p_1(i) \leftarrow 1/K$ for all $i = 1, \ldots, K$
    **for** $t = 1, \ldots, T$ **do**
        Select action $I_t \sim p_t$ and incur loss $\ell_t(I_t)$
        Observe losses $\{(i, \ell_t(i)) : i \in N_t(I_t)\}$ and graph $G_t$
        Construct a loss estimate $\widehat{\ell}_t$ for $\ell_t$                  $\triangleright$ e.g., (1) or (6)
        Let $p_{t+1} \leftarrow \arg\min_{p \in \Delta_K} \eta\langle \sum_{s=1}^{t} \widehat{\ell}_s, p \rangle + \psi_q(p)$
    **end for**

---

A key part of the standard regret analysis of $q$-FTRL (see, e.g., the proof of Lemma 3 in Appendix A) is handling the variance term, which, with the choice of estimator given in (1), takes the following form

$$B_t(q) = \sum_{i \in V} \frac{p_t(i)^{2-q}}{P_t(i)} \ . \tag{2}$$

By Hölder's inequality, this term can be immediately upper bounded by

$$B_t(q) \le \sum_{i \in V} p_t(i)^{1-q} \le \left( \sum_{i \in V} p_t(i) \right)^{1-q} \left( \sum_{i \in V} 1^{1/q} \right)^{q} = K^q \ ,$$

while previous results on the regret analysis of multiarmed bandits with graph feedback [29, 3] would give

$$B_t(q) \le \sum_{i \in V} \frac{p_t(i)}{P_t(i)} \le \alpha \ .$$

However, the former result would only recover a $\mathcal{O}(\sqrt{KT})$ regret bound (regardless of $\alpha$) with the best choice of $q = 1/2$, which could be trivially achieved by ignoring side-observations of the losses, whereas the latter bound would only manage to achieve a $\mathcal{O}(\sqrt{\alpha T \ln K})$ regret bound, incurring the extra $\sqrt{\ln K}$ factor for all values of $\alpha$. Other results in the literature (e.g., see [2, 4, 16, 19, 22, 25, 32, 35]) do not bring an improvement in this setting when bounding the $B_t(q)$ term and, hence, do not suffice for achieving the desired regret bound. The following lemma provides a novel and improved bound on quantities of the same form as $B_t(q)$ in terms of the independence number $\alpha_t = \alpha$ of the undirected graph $G_t$.

**Lemma 1.** *Let $G = (V, E)$ be any undirected graph with $|V| = K$ vertices and independence number $\alpha(G) = \alpha$. Let $b \in [0,1]$, $p \in \Delta_K$ and consider any nonempty subset $U \subseteq \{v \in V : v \in N_G(v)\}$. Then,*

$$\sum_{v \in U} \frac{p(v)^{1+b}}{\sum_{u \in N_G(v)} p(u)} \le \alpha^{1-b} \ .$$

*Proof.* First of all, observe that we can restrict ourselves to the subgraph $G[U]$ induced by $U$, i.e., the graph $G[U] = (U, E \cap (U \times U))$. This is because the neighbourhoods in this graph are such that $N_{G[U]}(v) \subseteq N_G(v)$ for all $v \in U$, and its independence number is $\alpha(G[U]) \le \alpha(G)$. Hence, it suffices to prove the claimed inequality for any undirected graph $G = (V, E)$ with all self-loops, any $p \in [0,1]^K$ such that $\|p\|_1 \le 1$, and the choice $U = V$. We assume this in what follows without loss of generality.

For any subgraph $H \subseteq G$ with vertices $V(H) \subseteq V$, denote the quantity we want to upper bound by

$$Q(H) = \sum_{v \in V(H)} \frac{p(v)^{1+b}}{\sum_{u \in N_G(v)} p(u)} \ .$$

Our aim is thus to provide an upper bound to $Q(G)$.

Consider a greedy algorithm that incrementally constructs a subset of vertices in the following way: at each step, it selects a vertex $v$ that maximizes $p(v)^b / \left( \sum_{u \in N_G(v)} p(u) \right)$, it adds $v$ to the solution, and it removes $v$ from $G$ together with its neighbourhood $N_G(v)$. This step is iterated on the remaining graph until no vertex is left.

Let $S = \{v_1, \ldots, v_s\} \subseteq V$ be the solution returned by the above greedy algorithm on $G$. Also let $G_1, \ldots, G_{s+1}$ be the sequence of graphs induced by the operations of the algorithm, where $G_1 = G$ and $G_{s+1}$ is the empty graph, and let $N_r(v) = N_{G_r}(v)$ for $v \in V(G_r)$. At every step $r \in [s]$ of the greedy algorithm, the contribution to $Q(G)$ of the removed vertices $N_r(v_r)$ amounts to

$$Q(G_r) - Q(G_{r+1}) = \sum_{v \in N_r(v_r)} \frac{p(v)^{1+b}}{\sum_{u \in N_1(v)} p(u)} \leq \sum_{v \in N_r(v_r)} p(v) \frac{p(v_r)^b}{\sum_{u \in N_1(v_r)} p(u)}$$

$$\leq \frac{\sum_{v \in N_1(v_r)} p(v)}{\sum_{u \in N_1(v_r)} p(u)} p(v_r)^b = p(v_r)^b \ ,$$

where the last inequality is due to the fact that $N_i(v) \subseteq N_j(v)$ for all $i \geq j$ and $v \in V_i$. Therefore, we can observe that

$$Q(G) = \sum_{r=1}^{s} \big( Q(G_r) - Q(G_{r+1}) \big) \leq \sum_{v \in S} p(v)^b \ .$$

The solution $S$ is an independent set of $G$ by construction. Consider now any independent set $A \subseteq V$ of $G$. We have that

$$\sum_{v \in A} p(v)^b \leq \max_{x \in \Delta_K} \sum_{v \in A} x(v)^b = |A| \max_{x \in \Delta_K} \sum_{v \in A} \frac{x(v)^b}{|A|}$$

$$\leq |A| \max_{x \in \Delta_K} \left( \frac{1}{|A|} \sum_{v \in A} x(v) \right)^b \leq |A|^{1-b} \leq \alpha^{1-b} \ , \tag{3}$$

where the second inequality follows by Jensen's inequality and the fact that $b \in [0, 1]$. $\qquad \square$

Observe that this upper bound is tight for general probability distributions $p \in \Delta_K$ over the vertices $V$ of any strongly observable undirected graph $G$ (containing at least one self-loop), as it is exactly achieved by the distribution $p^\star \in \Delta_K$ defined as $p^\star(i) = \frac{1}{|S|} \mathbb{I} \{i \in S\}$ for some maximum independent set $S \subseteq V$ of $G$. Using this lemma, the following theorem provides our improved upper bound under the simplifying assumptions we made thus far.

**Theorem 1.** *Let $G_1, \ldots, G_T$ be a sequence of undirected feedback graphs, where each $G_t$ contains all self-loops and has independence number $\alpha_t = \alpha$ for some common value $\alpha \in [K]$. If Algorithm 1 is run with input*

$$q = \frac{1}{2} \left( 1 + \frac{\ln(K/\alpha)}{\sqrt{\ln(K/\alpha)^2 + 4} + 2} \right) \in [1/2, 1) \qquad and \qquad \eta = \sqrt{\frac{2qK^{1-q}}{T(1-q)\alpha^q}} \ ,$$

*and loss estimates (1), then its regret satisfies $R_T \leq 2\sqrt{e\alpha T \left( 2 + \ln(K/\alpha) \right)}$.*

*Proof.* One can verify that for any $i \in V$, the loss estimate $\widehat{\ell}_t(i)$ defined in (1) satisfies $\mathbb{E}_t \big[ \widehat{\ell}_t(i)^2 \big] \leq 1/P_t(i)$. Hence, using also that $\mathbb{E}_t \big[ \widehat{\ell}_t(i) \big] = \ell_t(i)$, Lemma 2 in Appendix A implies that

$$R_T \leq \frac{K^{1-q}}{\eta(1-q)} + \frac{\eta}{2q} \sum_{t=1}^{T} \mathbb{E} \left[ \sum_{i \in V} \frac{p_t(i)^{2-q}}{P_t(i)} \right] \tag{4}$$

$$\leq \frac{K^{1-q}}{\eta(1-q)} + \frac{\eta}{2q} \alpha^q T \ , \tag{5}$$

where the second inequality follows by Lemma 1 with $b = 1 - q$ since all actions $i \in V$ are such that $i \in N_G(i)$. Our choices for $q$ and $\eta$ allow us to further upper bound the right-hand side of (5) by

$$\sqrt{\frac{2K^{1-q}\alpha^q}{q(1-q)}T} = \sqrt{2T\exp\left(1 + \frac{1}{2}\ln(\alpha K) - \frac{1}{2}\sqrt{\ln(K/\alpha)^2 + 4}\right)\left(2 + \sqrt{\ln(K/\alpha)^2 + 4}\right)}$$

$$\leq \sqrt{2e\alpha T\left(2 + \sqrt{\ln(K/\alpha)^2 + 4}\right)} \leq 2\sqrt{e\alpha T\sqrt{\ln(K/\alpha)^2 + 4}}$$

$$\leq 2\sqrt{e\alpha T\left(2 + \ln(K/\alpha)\right)} \ . \qquad \qquad \square$$

The regret bound achieved in the above theorem achieves the optimal regret bound for the experts setting (i.e., $\alpha = 1$) and the bandits setting (i.e., $\alpha = K$) simultaneously. Moreover, it interpolates the intermediate cases for $\alpha$ ranging between $1$ and $K$, introducing the multiplicative logarithmic factor only for graphs with independence number strictly smaller than $K$. We remark that the chosen values of $q$ and $\eta$ do in fact minimize the right-hand side of (5). Note that we relied on the knowledge of $\alpha$ to tune the parameter $q$. This is undesirable in general. We will show how to lift this requirement in Section 4. The same comment applies to Theorem 2, below.

We now show how to achieve the improved regret bound of Theorem 1 in the case of strongly observable undirected feedback graphs where some self-loops may be missing; i.e., there may be actions $i \in V$ such that $i \notin N_G(i)$. Using the loss estimator defined in (1) may lead to a large variance term due to the presence of actions without self-loops. One approach to deal with this—see, e.g., [35] or [28]—is to suitably alter the loss estimates of these actions.

Define $S_t = \{i \in V : i \notin N_t(i)\}$ as the subset of actions without self-loops in the feedback graph $G_t$ at each time step $t \in [T]$. The idea is that we need to carefully handle some action $i \in S_t$ only in the case when the probability $p_t(i)$ of choosing $i$ at round $t$ is sufficiently large, say, larger than $1/2$. Define the set of such actions as $J_t = \{i \in S_t : p_t(i) > 1/2\}$ and observe that $|J_t| \leq 1$. Similarly to [35], define new loss estimates

$$\widehat{\ell}_t(i) = \begin{cases} \frac{\ell_t(i)}{P_t(i)}\mathbb{I}\{I_t \in N_t(i)\} & \text{if } i \in V \setminus J_t \\ \frac{\ell_t(i)-1}{P_t(i)}\mathbb{I}\{I_t \in N_t(i)\} + 1 & \text{if } i \in J_t \end{cases} \qquad (6)$$

for which it still holds that $\mathbb{E}_t[\widehat{\ell}_t] = \ell_t$ and that $\mathbb{E}_t[\widehat{\ell}_t(i)^2] \leq 1/P_t(i)$ for all $i \notin J_t$. This change, along with the use of Lemma 1 for the actions in $V \setminus S_t$, suffices in order to prove the following regret bound (see Appendix B for the proof) when the feedback graphs do not necessarily contain self-loops for all actions.

**Theorem 2.** *Let $G_1, \ldots, G_T$ be a sequence of strongly observable undirected feedback graphs, where each $G_t$ has independence number $\alpha_t = \alpha$ for some common value $\alpha \in [K]$. If Algorithm 1 is run with input*

$$q = \frac{1}{2}\left(1 + \frac{\ln(K/\alpha)}{\sqrt{\ln(K/\alpha)^2 + 4} + 2}\right) \in [1/2, 1) \qquad \text{and} \qquad \eta = \frac{1}{3}\sqrt{\frac{2qK^{1-q}}{T(1-q)\alpha^q}} \ ,$$

*and loss estimates (6), then its regret satisfies $R_T \leq 6\sqrt{e\alpha T\left(2 + \ln(K/\alpha)\right)}$.*

## 4 Adapting to Arbitrary Sequences of Graphs

In the previous section, we assumed for simplicity that all the graphs have the same independence number. This independence number was then used to tune $q$, the parameter of the Tsallis entropy regularizer used by the algorithm. In this section, we show how to extend our approach to the case when the independence numbers of the graphs are neither the same nor known a-priori by the learner. Had these independence numbers been known a-priori, one approach is to set $q$ as in Theorem 2 but using the average independence number

$$\bar{\alpha}_T = \frac{1}{T}\sum_{t=1}^{T}\alpha_t \ .$$

Doing so would allow us to achieve a $\mathcal{O}\left(\sqrt{\sum_{t=1}^{T} \alpha_t(1 + \ln(K/\bar{\alpha}_T))}\right)$ regret bound. We now show that we can still recover a bound of the same order without prior knowledge of $\bar{\alpha}_T$. For round $t$ and any fixed $q \in [0, 1]$, define

$$H_t(q) = \sum_{i \in V \setminus S_t} \frac{p_t(i)^{2-q}}{P_t(i)} \ .$$

We know from Lemma 1 that $H_t(q) \leq \alpha_t^q$. Thus, we can leverage these observations and use a doubling trick (similar in principle to [3]) to guess the value of $\bar{\alpha}_T$. This approach is outlined in Algorithm 2. Starting with $r = 0$ and $T_r = 1$, the idea is to instantiate Algorithm 1 at time-step $T_r$ with $q$ and $\eta$ set as in Theorem 2 but with $2^r$ replacing the independence number. Then, at $t \geq T_r$, we increment $r$ and restart Algorithm 1 only if

$$\frac{1}{T} \sum_{s=T_r}^{t} H_s(q_r)^{1/q_r} > 2^{r+1},$$

since (again thanks to Lemma 1) the left-hand side of the above inequality is always upper bounded by $\bar{\alpha}_T$. The following theorem shows that this approach essentially enjoys the same regret bound of Theorem 2 up to an additive $\log_2 \bar{\alpha}_T$ term.

---

**Algorithm 2** $q$-FTRL for an arbitrary sequence of strongly observable undirected graphs

    **input:** Time horizon $T$
    **define:** For each $r \in \{0, \ldots, \lfloor \log_2 K \rfloor\}$,

$$q_r = \frac{1}{2}\left(1 + \frac{\ln(K/2^r)}{\sqrt{\ln(K/2^r)^2 + 4} + 2}\right) \qquad \text{and} \qquad \eta_r = \sqrt{\frac{2q_r K^{1-q_r}}{11T(1-q_r)(2^r)^{q_r}}}$$

    **initialization:** $T_0 \leftarrow 1$, $r \leftarrow 0$, instantiate Algorithm 1 with $q = q_0$, $\eta = \eta_0$, and loss estimates (6)
    **for** $t = 1, \ldots, T$ **do**
        Perform one step of the current instance of Algorithm 1
        **if** $\frac{1}{T} \sum_{s=T_r}^{t} H_s(q_r)^{1/q_r} > 2^{r+1}$ **then**
            $r \leftarrow r + 1$
            $T_r \leftarrow t + 1$
            Restart Algorithm 1 with $q = q_r$, $\eta = \eta_r$, and loss estimates (6)
        **end if**
    **end for**

---

**Theorem 3.** *Let $C = 4\sqrt{6}e^{\frac{\sqrt{\pi}+\sqrt{4-2\ln 2}}{\ln 2}}$. Then, the regret of Algorithm 2 satisfies*

$$R_T \leq C\sqrt{\sum_{t=1}^{T} \alpha_t\left(2 + \ln\left(\frac{K}{\bar{\alpha}_T}\right)\right)} + \log_2 \bar{\alpha}_T \ .$$

*Proof sketch.* For simplicity, we sketch here the proof for the case when in every round $t$, all the nodes have self-loops; hence, $H_t(q) = B_t(q)$. See the full proof in Appendix C, which treats the general case in a similar manner. Let $n = \lceil \log_2 \bar{\alpha}_T \rceil$ and assume without loss of generality that $\bar{\alpha}_T > 1$. Since Lemma 1 implies that for any $r$ and $t$, $B_t(q_r) \leq \alpha_t^{q_r}$, we have as a consequence that for any $t \geq T_r$,

$$\frac{1}{T} \sum_{s=T_r}^{t} B_s(q_r)^{1/q_r} \leq \frac{1}{T} \sum_{s=T_r}^{t} \alpha_s \leq \bar{\alpha}_T \leq 2^n \ .$$

Hence, the maximum value of $r$ that the algorithm can reach is $n - 1$. In doing so, we will execute $n$ instances of Algorithm 1, each corresponding to a value of $r \in \{0, \ldots, n-1\}$. For every such $r$, we upper bound the instantaneous regret at step $T_{r+1} - 1$ (the step when the restarting condition is satisfied) by 1, hence the added $\log_2 \bar{\alpha}_T$ term in the regret bound. For the rest of the interval; namely, for $t \in [T_r, T_{r+1} - 2]$, we have via (4) that the regret of Algorithm 1 is bounded by

$$\frac{K^{1-q_r}}{\eta_r(1-q_r)} + \frac{\eta_r}{2q_r} \mathbb{E} \sum_{t=T_r}^{T_{r+1}-2} B_t(q_r) \ . \tag{7}$$

Define $T_{r:r+1} = T_{r+1} - T_r - 1$, and notice that

$$\sum_{t=T_r}^{T_{r+1}-2} B_t(q_r) \leq T_{r:r+1} \left( \frac{1}{T_{r:r+1}} \sum_{t=T_r}^{T_{r+1}-2} B_t(q_r)^{1/q_r} \right)^{q_r}$$

$$\leq T_{r:r+1} \left( \frac{T}{T_{r:r+1}} 2^{r+1} \right)^{q_r} \leq 2T \left( 2^r \right)^{q_r} ,$$

where the first inequality follows due to Jensen's inequality since $q_r \in (0, 1)$, and the second follows from the restarting condition of Algorithm 2. After, plugging this back into (7), we can simply use the definitions of $\eta_r$ and $q_r$ and bound the resulting expression in a similar manner to the proof of Theorem 1. Overall, we get that

$$R_T \leq 4\sqrt{3eT} \sum_{r=0}^{n-1} \sqrt{2^r \ln\left(e^2 K 2^{-r}\right)} + \log_2 \bar{\alpha}_T ,$$

from which the theorem follows by using Lemma 4 in Appendix A, which shows, roughly speaking, that the sum on the right-hand side is of the same order as its last term. $\qquad\square$

Although Algorithm 2 requires knowledge of the time horizon, this can be dealt with by applying a standard doubling trick on $T$ at the cost of a larger constant factor. It is also noteworthy that the bound we obtained is of the form $\sqrt{T\bar{\alpha}_T(1 + \ln(K/\bar{\alpha}_T))}$ and not $\sqrt{\sum_t \alpha_t(1 + \ln(K/\alpha_t))}$. Although both coincide with the bound of Theorem 2 when $\alpha_t$ is the same for all time steps, the latter is smaller via the concavity of $x(1 + \ln(K/x))$ in $x$. It is not clear, however, whether there is a tuning of $q \in (0, 1)$ that can achieve the second bound (even with prior knowledge of the entire sequence $\alpha_1, \ldots, \alpha_T$ of independence numbers).

## 5 An Improved Lower Bound via Multitask Learning

In this section we provide a new lower bound on the minimax regret showing that, apart from the bandits case, a logarithmic factor is indeed necessary in general. When the graph is fixed over time, it is known that a lower bound of order $\sqrt{\alpha T}$ holds for any value of $\alpha$ [3, 29]. Whereas for the experts case ($\alpha = 1$), the minimax regret is of order[3] $\sqrt{T \ln K}$ [9]. The following theorem provides, for the first time, a lower bound that interpolates between the two aforementioned bounds for the intermediate values of $\alpha$.

**Theorem 4.** *Pick any $K \geq 2$ and any $\alpha$ such that $2 \leq \alpha \leq K$. Then, for any algorithm and for all $T \geq \frac{\alpha \log_\alpha K}{4 \log(4/3)}$, there exists a sequence of losses and feedback graphs $G_1, \ldots, G_T$ such that $\alpha(G_t) = \alpha$ for all $t = 1, \ldots, T$ and*

$$R_T \geq \frac{1}{18\sqrt{2}} \sqrt{\alpha T \log_\alpha K}.$$

In essence, the proof of this theorem (see Appendix D) constructs a sequence of feedback graphs and losses that is equivalent to a hard instance of the multitask bandit problem (MTB) [12], an important special case of combinatorial bandits with a convenient structure for proving lower bounds [6, 15, 21]. We consider a variant of MTB in which, at the beginning of each round, the decision-maker selects an arm to play in each one of $M$ stochastic bandit games. Subsequently, the decision-maker only observes (and suffers) the loss of the arm played in a single randomly selected game. For proving the lower bound, we use a class of stationary stochastic adversaries (i.e., environments), each generating graphs and losses in a manner that simulates an MTB instance.

Fix $2 \leq \alpha \leq K = |V|$ and assume for simplicity that $M = \log_\alpha K$ is an integer. We now construct an instance of online learning with time-varying feedback graphs $G_t = (V, E_t)$ with $\alpha(G_t) = \alpha$ that is equivalent to an MTB instance with $M$ bandit games each containing $\alpha$ "base actions". Since $K = \alpha^M$, we can uniquely identify each action in $V$ with a vector $a = \left(a(1), \ldots, a(M)\right)$ in

---

[3]As a lower bound, this is known to hold asymptotically as $K$ and $T$ grow. However, it can also be shown to hold non-asymptotically (though with worse leading constants); see [20, Theorem 3.22] or [11, Theorem 3.6].

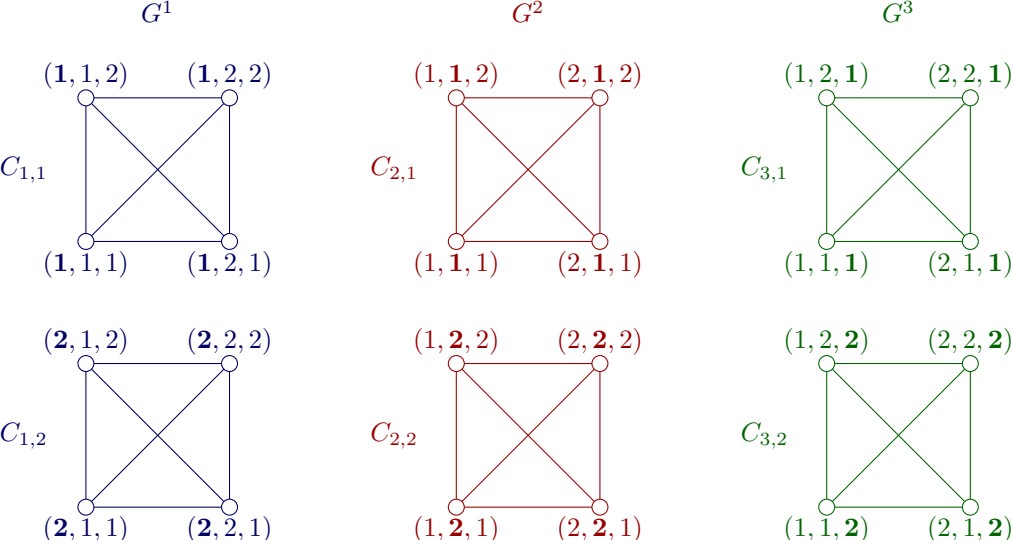

Figure 1: This figure shows an example of the multi-task bandit construction used to prove the lower bound. Here, $K = 8$ and $\alpha = 2$; thus, the number of games is $M = 3$. Each action is identified by a tuple of three numbers, each corresponding to a choice of one out of a pair of "base actions" in each game. Each of the three graphs in the figure corresponds to a game, such that two actions share an edge if and only if they choose the same base action in the corresponding game. At every round, a graph is randomly drawn, and all actions belonging to the same clique suffer the same loss.

$[\alpha]^M$. The action $a_t \in V$ chosen by the learner at round $t$ is equivalent to a choice of base actions $a_t(1), \ldots, a_t(M)$ in the $M$ games. The feedback graph at every round is sampled uniformly at random from a set of $M$ undirected graphs $\{G^i\}_{i=1}^M$, where $G^i = (V, E^i)$ is such that $(a, a') \in E^i$ if and only if $a(i) = a'(i)$. This means (see Figure 1) that each graph $G^i$ consists of $\alpha$ isolated cliques $\{C_{i,j}\}_{j=1}^\alpha$ such that an action $a$ belongs to clique $C_{i,j}$ if and only if $a(i) = j$. Clearly, the independence number of any such graph is $\alpha$. Drawing feedback graph $G_t = G^i$ corresponds to the activation of game $i$ in the MTB instance. Hence, choosing $a_t \in V$ with feedback graph $G_t = G^i$ is equivalent to playing base action $a_t(i)$ in game $i$ in the MTB. As for the losses, we enforce that, given a feedback graph $G_t$, all actions that belong to the same clique of the feedback graph are assigned the same loss. Namely, if $G_t = G^i$ and $a(i) = a'(i) = j$, then $\ell_t(a) = \ell_t(a')$, which can be seen as the loss $\ell_t(j)$ assigned to base action $j$ in game $G^i$. To choose the distribution of the losses for the base actions, we apply the classic needle-in-a-haystack approach of [7] over the $M$ games. More precisely, we construct a different environment for each action $a \in V$ in such a way that the distribution of the losses in each MTB game slightly favors (with a difference of a small $\varepsilon > 0$) the base action corresponding to $a$ in that game. The proof then proceeds similarly to, for example, the proof of Theorem 5 in [6] or Theorem 7 in [17].

While both our upper and lower bounds achieve the desired goal of interpolating between the minimax rates of experts and bandits, the logarithmic factors in the two bounds are not exactly matching. In particular, if we compare $1 + \log_2(K/\alpha)$ and $\log_\alpha K$, we can see that although they coincide at $\alpha = 2$ and $\alpha = K$, the former is larger for intermediate values. It is reasonable to believe that the upper bound is of the correct order, seeing as it arose naturally as a result of choosing the best parameter for the Tsallis entropy regularizer, whereas achieving the extra logarithmic term in the lower bound required a somewhat contrived construction.

## 6   Conclusions and Future Work

In this work, we have shown that a proper tuning of the $q$-FTRL algorithm allows one to achieve a $\mathcal{O}\big(\sqrt{\alpha T(1 + \ln(K/\alpha))}\big)$ regret for the problem of online learning with undirected strongly observable feedback graphs. Our bound interpolates between the minimax regret rates of the bandits and the experts problems, the two extremes of the strongly observable graph feedback spectrum.

Furthermore, we have shown that an analogous bound can be achieved when the graphs vary over time, and without requiring any prior knowledge on the graphs. These results are complemented by our new lower bound of $\Omega\big(\sqrt{\alpha T(\ln K)/(\ln \alpha)}\big)$, which holds for $\alpha \geq 2$ and shows the necessity of a logarithmic factor in the minimax regret except for the bandits case. While our results provide the tightest characterization to date of the minimax rate for this setting, closing the small remaining gap (likely on the lower bound side) is an interesting problem. After the submission of this manuscript, a subsequent work [14] showed a lower bound for fixed feedback graphs composed of disjoint cliques that would imply worst-case optimality (up to constant factors) of our proposed algorithm for each pair of $K$ and $\alpha$—see Appendix E for a more detailed comparison with results therein. Extending our results to the case of directed strongly observable feedback graphs is a considerably harder task—see Appendix F for a preliminary discussion. Better understanding this more general setting is an interesting future direction.

## Acknowledgements

KE, EE, and NCB gratefully acknowledge the financial support from the MUR PRIN grant 2022EKNE5K (Learning in Markets and Society), funded by the NextGenerationEU program within the PNRR scheme (M4C2, investment 1.1), the FAIR (Future Artificial Intelligence Research) project, funded by the NextGenerationEU program within the PNRR-PE-AI scheme (M4C2, investment 1.3, line on Artificial Intelligence), and the EU Horizon CL4-2022-HUMAN-02 research and innovation action under grant agreement 101120237, project ELIAS (European Lighthouse of AI for Sustainability). TC gratefully acknowledges the support of the University of Ottawa through grant GR002837 (Start-Up Funds) and that of the Natural Sciences and Engineering Research Council of Canada (NSERC) through grants RGPIN-2023-03688 (Discovery Grants Program) and DGECR-2023-00208 (Discovery Grants Program, DGECR - Discovery Launch Supplement).

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
