## A Auxiliary Results

**Lemma 2.** *If Algorithm 1 is run with $q \in (0, 1)$, learning rate $\eta > 0$, and non-negative loss estimates that satisfy $\mathbb{E}_t\big[\widehat{\ell}_t\big] = \ell_t$ for all $t = 1, \dots, T$, then its regret satisfies*

$$R_T \leq \frac{K^{1-q}}{(1-q)\eta} + \frac{\eta}{2q} \sum_{t=1}^{T} \mathbb{E}\left[\sum_{i \in V} p_t(i)^{2-q}\,\widehat{\ell}_t(i)^2\right] .$$

*Proof.* Let $i^* \in \arg\min_{i \in V} \sum_{t=1}^{T} \ell_t(i)$ be an action that minimizes the cumulative loss, and let $\mathbf{e}_{i^*} \in \mathbb{R}^K$ be an indicator vector for $i^*$. Recall that for $t \in [T]$, $\mathbb{E}_t[\cdot] = \mathbb{E}[\cdot \mid I_1, \dots, I_{t-1}]$, and notice that $p_t$ is measurable with respect to the $\sigma$-algebra generated by $I_1, \dots, I_{t-1}$. Hence, using that

$$\mathbb{E}_t\big[\ell_t(I_t)\big] = \sum_{i \in V} p_t(i)\ell_t(i) \qquad \text{and} \qquad \mathbb{E}_t\big[\widehat{\ell}_t\big] = \ell_t ,$$

we have, via the tower rule and the linearity of expectation, that

$$R_T = \mathbb{E}\left[\sum_{t=1}^{T} \ell_t(I_t)\right] - \sum_{t=1}^{T} \ell_t(i^*) = \mathbb{E}\left[\sum_{t=1}^{T}\langle p_t - \mathbf{e}_{i^*}, \ell_t\rangle\right] = \mathbb{E}\left[\sum_{t=1}^{T}\langle p_t - \mathbf{e}_{i^*}, \widehat{\ell}_t\rangle\right],$$

from which we can obtain the desired result by using Lemma 3 (which holds even if the loss $\widehat{\ell}_t$ at each round $t \in [T]$ depends on the prediction $p_t$ made at that round). $\qquad\square$

**Lemma 3.** *Let $q \in (0, 1)$, $\eta > 0$, and $(y_t)_{t=1}^{T}$ be an arbitrary sequence of non-negative loss vectors in $\mathbb{R}^K$. Let $(p_t)_{t=1}^{T+1}$ be the predictions of FTRL with decision set $\Delta_K$ and the q-Tsallis regularizer $\psi_q$ over this sequence of losses. That is, $p_1 = \arg\min_{p \in \Delta_K} \psi_q(p)$, and for $t \in [T]$,*

$$p_{t+1} = \arg\min_{p \in \Delta_K} \eta \sum_{s=1}^{t}\langle y_s, p\rangle + \psi_q(p) .$$

*Then for any $u \in \Delta_K$,*

$$\sum_{t=1}^{T}\langle p_t - u, y_t\rangle \leq \frac{K^{1-q}}{(1-q)\eta} + \frac{\eta}{2q} \sum_{t=1}^{T}\sum_{i \in V} p_t(i)^{2-q}\,y_t(i)^2 .$$

*Proof.* By Theorem 28.5 in [27], we have that

$$\begin{aligned}
\sum_{t=1}^{T}\langle p_t - u, y_t\rangle &\leq \frac{\psi_q(u) - \psi_q(p_1)}{\eta} + \sum_{t=1}^{T}\left(\langle p_t - p_{t+1}, y_t\rangle - \frac{1}{\eta}D_{\psi_q}(p_{t+1}, p_t)\right) \\
&= \frac{K^{1-q} - 1}{(1-q)\eta} + \sum_{t=1}^{T}\left(\langle p_t - p_{t+1}, y_t\rangle - \frac{1}{\eta}D_{\psi_q}(p_{t+1}, p_t)\right) \\
&\leq \frac{K^{1-q}}{(1-q)\eta} + \sum_{t=1}^{T}\left(\langle p_t - p_{t+1}, y_t\rangle - \frac{1}{\eta}D_{\psi_q}(p_{t+1}, p_t)\right) ,
\end{aligned}$$

where $D_{\psi_q}(\cdot, \cdot)$ is the Bregman divergence based on $\psi_q$. For bounding each summand in the second term, we follow a similar argument to that used in Theorem 30.2 in [27]. Namely, for each $i \in V$ and

round $t \in [T]$, define $\overline{y}_t(i) = \mathbb{I}\{p_{t+1}(i) \leq p_t(i)\}y_t(i)$. We then have that

$$\langle p_t - p_{t+1}, y_t \rangle - \frac{1}{\eta}D_{\psi_q}(p_{t+1}, p_t)$$

$$\leq \langle p_t - p_{t+1}, \overline{y}_t \rangle - \frac{1}{\eta}D_{\psi_q}(p_{t+1}, p_t)$$

$$= \frac{1}{\eta}\langle p_t - p_{t+1}, \eta\overline{y}_t \rangle - \frac{1}{2\eta}\big\|p_{t+1} - p_t\big\|^2_{\nabla^2\psi_q(z_t)}$$

$$\leq \frac{\eta}{2}\big\|\overline{y}_t\big\|^2_{(\nabla^2\psi_q(z_t))^{-1}}$$

$$= \frac{\eta}{2q}\sum_{i \in V} z_t(i)^{2-q}\,\overline{y}_t(i)^2$$

$$= \frac{\eta}{2q}\sum_{i \in V}\big(\gamma_t p_{t+1}(i) + (1-\gamma_t)p_t(i)\big)^{2-q}\,\overline{y}_t(i)^2$$

$$\leq \frac{\eta}{2q}\sum_{i \in V}p_t(i)^{2-q}\,\overline{y}_t(i)^2 + \gamma_t\frac{\eta}{2q}\sum_{i \in V}\big(p_{t+1}(i)^{2-q} - p_t(i)^{2-q}\big)\,\overline{y}_t(i)^2$$

$$\leq \frac{\eta}{2q}\sum_{i \in V}p_t(i)^{2-q}\,\overline{y}_t(i)^2$$

$$\leq \frac{\eta}{2q}\sum_{i \in V}p_t(i)^{2-q}\,y_t(i)^2 \ ,$$

where $z_t = \gamma_t p_{t+1} + (1-\gamma_t)p_t$ for some $\gamma_t \in [0,1]$; the first inequality holds due to the non-negativity of the losses, the second inequality is an application of the Fenchel-Young inequality, the second equality holds since the Hessian of $\psi_q$ is a diagonal matrix with $(\nabla^2\psi_q(x))_{i,i} = qx(i)^{q-2}$, the third inequality is an application of Jensen's inequality (since $q \in (0,1)$), and the fourth inequality holds since $\overline{y}_t(i) = 0$ for any $i$ such that $p_{t+1}(i)^{2-q} > p_t(i)^{2-q}$. $\qquad\square$

**Lemma 4.** *Let $a$ and $b$ be positive integers such that $2 \leq a \leq b$, and let $n = \lceil \log_2 a \rceil$. Then,*

$$\sum_{r=0}^{n-1}\sqrt{2^r\ln\big(e^2b2^{-r}\big)} \leq \frac{\sqrt{2\pi} + 2\sqrt{2-\ln 2}}{\ln 2}\sqrt{a\ln\left(\frac{e^2b}{a}\right)} \ .$$

*Proof.* Since $n \leq \log_2(2b)$ and $2^r\ln\big(e^2b2^{-r}\big)$ is monotonically increasing in $r$ for $r \in [0, \log_2(eb)]$, we can bound the sum by an integral:

$$\sum_{r=0}^{n-1}\sqrt{2^r\ln\big(e^2b2^{-r}\big)} \leq \int_0^n\sqrt{2^r\ln\big(e^2b2^{-r}\big)}\,\mathrm{d}r \ .$$

We proceed via a change of variable; let $x = e^2b2^{-r}$, and note that $\mathrm{d}r = -\frac{\mathrm{d}x}{x\ln 2}$. We then have that

$$\int_0^n\sqrt{2^r\ln\big(e^2b2^{-r}\big)}\,\mathrm{d}r = \sqrt{e^2b}\int_0^n\sqrt{\frac{2^r}{e^2b}\ln\big(e^2b2^{-r}\big)}\,\mathrm{d}r$$

$$= -\frac{e\sqrt{b}}{\ln 2}\int_{e^2b}^{e^2b2^{-n}}\sqrt{\frac{\ln x}{x^3}}\,\mathrm{d}x = \frac{e\sqrt{b}}{\ln 2}\int_{e^2b2^{-n}}^{e^2b}\sqrt{\frac{\ln x}{x^3}}\,\mathrm{d}x$$

$$= \frac{e\sqrt{b}}{\ln 2}\left[-\sqrt{2\pi}\cdot\mathrm{erfc}\big(\sqrt{(\ln x)/2}\big) - 2\sqrt{(\ln x)/x}\right]_{e^2b2^{-n}}^{e^2b}$$

$$\leq \frac{e\sqrt{b}}{\ln 2}\left(\sqrt{2\pi}\cdot\mathrm{erfc}\big(\sqrt{\ln(e^2b2^{-n})/2}\big) + 2\sqrt{\frac{2^n\ln(e^2b2^{-n})}{e^2b}}\right) \ ,$$

where $\text{erfc}(x) = 1 - \frac{2}{\sqrt{\pi}} \int_0^x \exp(-z^2)\, dz$ is the complementary Gaussian error function, which is always positive. By [13, Theorem 1], we have that $\text{erfc}(x) \leq \exp(-x^2)$. Consequently,

$$\int_0^n \sqrt{2^r \ln\left(e^2 b 2^{-r}\right)}\, dr \leq \frac{e\sqrt{b}}{\ln 2} \left( \sqrt{2\pi}\sqrt{\frac{2^n}{e^2 b}} + 2\sqrt{\frac{2^n \ln(e^2 b 2^{-n})}{e^2 b}} \right)$$

$$= \frac{\sqrt{2^n}}{\ln 2} \left( \sqrt{2\pi} + 2\sqrt{\ln(e^2 b 2^{-n})} \right)$$

$$\leq \frac{\sqrt{2a}}{\ln 2} \left( \sqrt{2\pi} + 2\sqrt{\ln\left(\frac{e^2 b}{2a}\right)} \right)$$

$$\leq \frac{\sqrt{2\pi} + 2\sqrt{2 - \ln 2}}{\ln 2} \sqrt{a \ln\left(\frac{e^2 b}{a}\right)}\ ,$$

where in the second inequality we used once again the fact that $2^r \ln\left(e^2 b 2^{-r}\right)$ is monotonically increasing in $r$ for $r \in [0, \log_2(eb)]$ to replace $n$ with $\log_2(a) + 1$, and the last inequality holds since $b/a \geq 1$. $\qquad\square$

## B  Proofs of Section 3

In this section, we provide the proof of Theorem 2, which is restated below.

**Theorem 2.** *Let $G_1, \ldots, G_T$ be a sequence of strongly observable undirected feedback graphs, where each $G_t$ has independence number $\alpha_t = \alpha$ for some common value $\alpha \in [K]$. If Algorithm 1 is run with input*

$$q = \frac{1}{2}\left(1 + \frac{\ln(K/\alpha)}{\sqrt{\ln(K/\alpha)^2 + 4} + 2}\right) \in [1/2, 1) \qquad \text{and} \qquad \eta = \frac{1}{3}\sqrt{\frac{2qK^{1-q}}{T(1-q)\alpha^q}}\ ,$$

*and loss estimates* (6), *then its regret satisfies $R_T \leq 6\sqrt{e\alpha T\left(2 + \ln(K/\alpha)\right)}$.*

*Proof.* Let $i^* \in \arg\min_{i \in V} \sum_{t=1}^T \ell_t(i)$ and $\mathbf{e}_{i^*} \in \mathbb{R}^K$ be its indicator vector. Whenever $J_t$ is nonempty, let $j_t \in V$ be the only action such that $J_t = \{j_t\}$. Similarly to [35], let $z_t = \mathbb{I}\{J_t \neq \emptyset\}\mathbb{I}\{I_t \in N_t(j_t)\}\frac{1-\ell_t(j_t)}{1-p_t(j_t)}$ and define new losses $\widetilde{\ell}_t(i) = \widehat{\ell}_t(i) + z_t$ for each time step $t \in [T]$ and each action $i \in V$. Since $p_t, \mathbf{e}_{i^*} \in \Delta_K$, we have that $\langle p_t - \mathbf{e}_{i^*}, \widehat{\ell}_t \rangle = \langle p_t - \mathbf{e}_{i^*}, \widetilde{\ell}_t \rangle$ for every $t \in [T]$. Then, using the fact that $\mathbb{E}_t[\widehat{\ell}_t] = \ell_t$, we get that

$$R_T = \mathbb{E}\left[\sum_{t=1}^T \langle p_t - \mathbf{e}_{i^*}, \widehat{\ell}_t \rangle\right] = \mathbb{E}\left[\sum_{t=1}^T \langle p_t - \mathbf{e}_{i^*}, \widetilde{\ell}_t \rangle\right]\ ,$$

where the first equality holds via the same arguments made in the proof of Lemma 2. If we consider the optimization step of Algorithm 1, computing the same inner product over the new losses $\widetilde{\ell}_1, \ldots, \widetilde{\ell}_T$ for some $p \in \Delta_K$ gives

$$\left\langle \sum_{s=1}^t \widetilde{\ell}_s, p \right\rangle = \sum_{s=1}^t z_s + \left\langle \sum_{s=1}^t \widehat{\ell}_s, p \right\rangle\ ,$$

where the sum $\sum_{s=1}^t z_s$ is constant with respect to $p$. This implies that the objective functions in terms of either $(\widehat{\ell}_t)_{t \in [T]}$ and $(\widetilde{\ell}_t)_{t \in [T]}$, respectively, are minimized by the same probability distributions. However, notice that, unlike $(\widehat{\ell}_t)_{t \in [T]}$, the loss vectors in $(\widetilde{\ell}_t)_{t \in [T]}$ are always non-negative. Consequently, similar to the proof of Lemma 2, we may apply Lemma 3 to upper bound the regret of Algorithm 1 in terms of the losses $(\widetilde{\ell}_t)_{t \in [T]}$. Doing so gives

$$\mathbb{E}\left[\sum_{t=1}^T \langle p_t - \mathbf{e}_{i^*}, \widetilde{\ell}_t \rangle\right] \leq \frac{K^{1-q}}{\eta(1-q)} + \frac{\eta}{2q}\sum_{t=1}^T \mathbb{E}\left[\sum_{i \in V} p_t(i)^{2-q}\, \mathbb{E}_t\left[\widetilde{\ell}_t(i)^2\right]\right]\ . \tag{8}$$

We can bound the second term by observing that $\widetilde{\ell}_t(j_t) = 1$ whenever $J_t \neq \emptyset$. Therefore,

$$\sum_{i \in V} p_t(i)^{2-q} \, \mathbb{E}_t\left[\widetilde{\ell}_t(i)^2\right] \leq 2 \sum_{i \in V \setminus J_t} p_t(i)^{2-q} \, \mathbb{E}_t\left[\widehat{\ell}_t(i)^2\right] + 2\,\mathbb{E}_t\left[z_t^2\right] \sum_{i \in V \setminus J_t} p_t(i)^{2-q} + 1$$

$$\leq 2 \sum_{i \in V \setminus J_t} \frac{p_t(i)^{2-q}}{P_t(i)} + 2\,\mathbb{E}_t\left[z_t^2\right] \sum_{i \in V \setminus J_t} p_t(i)^{2-q} + 1$$

$$\leq 2 \sum_{i \in V \setminus J_t} \frac{p_t(i)^{2-q}}{P_t(i)} + 3 \ ,$$

where the second inequality holds because $\mathbb{E}_t\left[\widehat{\ell}_t(i)^2\right] \leq 1/P_t(i)$ for all $i \notin J_t$, and the third inequality follows from the fact that

$$\mathbb{E}_t\left[z_t^2\right] \sum_{i \in V \setminus J_t} p_t(i)^{2-q} = \mathbb{I}\left\{J_t \neq \emptyset\right\} \frac{\left(1 - \ell_t(j_t)\right)^2}{1 - p_t(j_t)} \sum_{i \in V \setminus J_t} p_t(i)^{2-q} \leq 1 \ .$$

We can handle the remaining sum by separating it over nodes $i \in S_t$, which satisfy $P_t(i) = 1 - p_t(i)$ because of strong observability, and those in $\overline{S}_t = V \setminus S_t$. In the first case, any node $i \in S_t \setminus J_t$ has $p_t(i) \leq 1/2$ and thus

$$\sum_{i \in S_t \setminus J_t} \frac{p_t(i)^{2-q}}{P_t(i)} = \sum_{i \in S_t \setminus J_t} \frac{p_t(i)^{2-q}}{1 - p_t(i)} \leq 2 \sum_{i \in S_t \setminus J_t} p_t(i)^{2-q} \leq 2 \ .$$

while in the second case we have that $\sum_{i \in \overline{S}_t} p_t(i)^{2-q}/P_t(i) \leq \alpha^q$ by Lemma 1 with $U = \overline{S}_t$ and $b = 1 - q$. Overall, we have shown that

$$\sum_{i \in V} p_t(i)^{2-q} \, \mathbb{E}_t\left[\widetilde{\ell}_t(i)^2\right] \leq 2 \sum_{i \in \overline{S}_t} \frac{p_t(i)^{2-q}}{P_t(i)} + 7 \leq 2\alpha^q + 7 \leq 9\alpha^q \ . \tag{9}$$

Plugging back into (8), we obtain that

$$R_T \leq \frac{K^{1-q}}{\eta(1-q)} + \frac{9\eta}{2q}\alpha^q T$$

$$= 3\sqrt{\frac{2K^{1-q}\alpha^q}{q(1-q)}T}$$

$$\leq 6\sqrt{e\alpha T \left(2 + \ln(K/\alpha)\right)} \ ,$$

where the equality is due to our choice of $\eta$, and the last inequality follows as in the proof of Theorem 1 together with our choice of $q$. $\qquad\square$

## C    Proofs of Section 4

In this section, we provide the proof of Theorem 3, which is restated below.

**Theorem 3.** *Let* $C = 4\sqrt{6e}\,\frac{\sqrt{\pi} + \sqrt{4 - 2\ln 2}}{\ln 2}$. *Then, the regret of Algorithm 2 satisfies*

$$R_T \leq C\sqrt{\sum_{t=1}^{T} \alpha_t \left(2 + \ln\left(\frac{K}{\bar{\alpha}_T}\right)\right)} + \log_2 \bar{\alpha}_T \ .$$

*Proof.* Notice that if $\bar{\alpha}_T = 1$, the initial guess is correct and the algorithm will never restart. Moreover, since in this case we have that $\alpha_t = 1$ for all $t$, the theorem follows trivially from the regret bound of Theorem 2. Hence, we can assume for what follows that $\bar{\alpha}_T > 1$. Let $i^* \in \arg\min_{i \in [K]} \sum_{t=1}^{T} \ell_t(i)$ and $n = \lceil \log_2 \bar{\alpha}_T \rceil$. Note that the maximum value of $r$ that the algorithm can reach is $n - 1$. To

see this, observe that Lemma 1 implies that for any $r$ and $t$, $H_t(q_r) \le \alpha_t^{q_r}$. Consequently, for any $t \ge T_r$,

$$\frac{1}{T} \sum_{s=T_r}^{t} H_s(q_r)^{1/q_r} \le \frac{1}{T} \sum_{s=T_r}^{t} \alpha_s \le \bar{\alpha}_T \le 2^n \ .$$

For $t \in [T]$, let $r_t$ be the value of $r$ at round $t$. Without loss of generality, we assume that $r$ takes each value in $\{0, \ldots, n-1\}$ for at least two rounds. Additionally, we define $T_n = T + 2$ for convenience. We start by decomposing the regret over the $n$ intervals (each corresponding to a value of $r$ in $\{0, \ldots, n-1\}$) and bounding the instantaneous regret with 1 for each step in which we restart (i.e., at the last step of each but the last interval):

$$R_T = \mathbb{E}\left[ \sum_{t=1}^{T} \left( \ell_t(I_t) - \ell_t(i^*) \right) \right]$$

$$\le \mathbb{E}\left[ \sum_{r=0}^{n-1} \sum_{t=T_r}^{T_{r+1}-2} \left( \ell_t(I_t) - \ell_t(i^*) \right) \right] + n - 1$$

$$\le \mathbb{E}\left[ \sum_{r=0}^{n-1} \sum_{t=T_r}^{T_{r+1}-2} \left( \ell_t(I_t) - \ell_t(i^*) \right) \right] + \log_2 \bar{\alpha}_T \ . \tag{10}$$

For what follows, let $\mathbf{e}_{i^*} \in \mathbb{R}^K$ be an indicator vector for $i^*$ and let $\widetilde{\ell}_t$ be as defined in the proof of Theorem 2. Fix $r \in \{0, \ldots, n-1\}$, we proceed by bounding the regret in the interval $[T_r, T_{r+1} - 2]$:

$$\mathbb{E}\left[ \sum_{t=T_r}^{T_{r+1}-2} \left( \ell_t(I_t) - \ell_t(i^*) \right) \right]$$

$$= \mathbb{E}\left[ \sum_{t=1}^{T} \mathbb{I}\left\{ r_t = r, \frac{1}{T} \sum_{s=T_{r_t}}^{t} H_s(q_{r_t})^{1/q_{r_t}} \le 2^{r_t + 1} \right\} \left( \ell_t(I_t) - \ell_t(i^*) \right) \right]$$

$$\overset{(a)}{=} \mathbb{E}\left[ \sum_{t=1}^{T} \mathbb{I}\left\{ r_t = r, \frac{1}{T} \sum_{s=T_{r_t}}^{t} H_s(q_{r_t})^{1/q_{r_t}} \le 2^{r_t + 1} \right\} \langle p_t - \mathbf{e}_{i^*}, \widehat{\ell}_t \rangle \right]$$

$$\overset{(b)}{=} \mathbb{E}\left[ \sum_{t=1}^{T} \mathbb{I}\left\{ r_t = r, \frac{1}{T} \sum_{s=T_{r_t}}^{t} H_s(q_{r_t})^{1/q_{r_t}} \le 2^{r_t + 1} \right\} \langle p_t - \mathbf{e}_{i^*}, \widetilde{\ell}_t \rangle \right]$$

$$= \mathbb{E}\left[ \sum_{t=T_r}^{T_{r+1}-2} \langle p_t - \mathbf{e}_{i^*}, \widetilde{\ell}_t \rangle \right]$$

$$\overset{(c)}{\le} \frac{K^{1-q_r}}{\eta_r(1 - q_r)} + \frac{\eta_r}{2 q_r} \mathbb{E}\left[ \sum_{t=T_r}^{T_{r+1}-2} \sum_{i \in V} p_t(i)^{2-q_r} \widetilde{\ell}_t(i)^2 \right]$$

$$\overset{(d)}{=} \frac{K^{1-q_r}}{\eta_r(1 - q_r)} + \frac{\eta_r}{2 q_r} \mathbb{E}\left[ \sum_{t=1}^{T} \mathbb{I}\left\{ r_t = r, \frac{1}{T} \sum_{s=T_{r_t}}^{t} H_s(q_{r_t})^{1/q_{r_t}} \le 2^{r_t + 1} \right\} \mathbb{E}_t\left[ \sum_{i \in V} p_t(i)^{2-q_r} \widetilde{\ell}_t(i)^2 \right] \right]$$

$$\overset{(e)}{\le} \frac{K^{1-q_r}}{\eta_r(1 - q_r)} + \frac{\eta_r}{2 q_r} \mathbb{E}\left[ \sum_{t=1}^{T} \mathbb{I}\left\{ r_t = r, \frac{1}{T} \sum_{s=T_{r_t}}^{t} H_s(q_{r_t})^{1/q_{r_t}} \le 2^{r_t + 1} \right\} (2 H_t(q_r) + 7) \right]$$

$$= \frac{K^{1-q_r}}{\eta_r(1 - q_r)} + \frac{\eta_r}{2 q_r} \mathbb{E}\left[ \sum_{t=T_r}^{T_{r+1}-2} (2 H_t(q_r) + 7) \right] \ , \tag{11}$$

where $(a)$ follows since $\mathbb{E}_t[\ell_t(I_t)] = \sum_{i \in V} p_t(i) \ell_t(i)$, $\mathbb{E}_t[\widehat{\ell}_t] = \ell_t$, and the indicator at round $t$ is measurable with respect to $\sigma(I_1, \ldots, I_{t-1})$, that is, the $\sigma$-algebra generated by $I_1, \ldots, I_{t-1}$; $(b)$ follows since $\langle p_t - \mathbf{e}_{i^*}, \widehat{\ell}_t \rangle = \langle p_t - \mathbf{e}_{i^*}, \widetilde{\ell}_t \rangle$ holds by the definition of $\widetilde{\ell}_t$; $(c)$ is an application of Lemma 3, justifiable with the same argument leading to (8) in the proof of Theorem 2; $(d)$ uses once

again that the indicator at round $t$ is measurable with respect to $\sigma(I_1, \ldots, I_{t-1})$; finally, $(e)$ follows via (9). Define $T_{r:r+1} = T_{r+1} - T_r - 1$, and notice that

$$\sum_{t=T_r}^{T_{r+1}-2} H_t(q_r) = \frac{T_{r:r+1}}{T_{r:r+1}} \sum_{t=T_r}^{T_{r+1}-2} \left(H_t(q_r)^{1/q_r}\right)^{q_r}$$

$$\leq T_{r:r+1} \left(\frac{1}{T_{r:r+1}} \sum_{t=T_r}^{T_{r+1}-2} H_t(q_r)^{1/q_r}\right)^{q_r}$$

$$\leq T_{r:r+1} \left(\frac{T}{T_{r:r+1}} 2^{r+1}\right)^{q_r}$$

$$\leq 2T\left(2^r\right)^{q_r} ,$$

where the first inequality follows due to Jensen's inequality since $q_r \in (0,1)$, and the second follows from the restarting condition of Algorithm 2. Next, we plug this inequality back into (11), and then, similar to the proof of Theorem 2, we use the definitions of $\eta_r$ and $q_r$ and bound the resulting expression to get that

$$\mathbb{E}\left[\sum_{t=T_r}^{T_{r+1}-2} \left(\ell_t(I_t) - \ell_t(i^*)\right)\right] \leq \frac{K^{1-q_r}}{\eta_r(1-q_r)} + \frac{11\eta_r}{2q_r} T\left(2^r\right)^{q_r}$$

$$\leq 2\sqrt{11eT2^r\left(2 + \ln\left(K2^{-r}\right)\right)} \leq 4\sqrt{3eT2^r \ln\left(e^2 K 2^{-r}\right)} .$$

We then sum this quantity over $r$ and use Lemma 4 to get that

$$\mathbb{E}\left[\sum_{r=0}^{n-1} \sum_{t=T_r}^{T_{r+1}-2} \left(\ell_t(I_t) - \ell_t(i^*)\right)\right] \leq 4\sqrt{3eT} \sum_{r=0}^{n-1} \sqrt{2^r \ln\left(e^2 K 2^{-r}\right)}$$

$$\leq 4\sqrt{6e}\frac{\sqrt{\pi} + \sqrt{4-2\ln 2}}{\ln 2}\sqrt{\overline{\alpha}_T T\left(2 + \ln\left(K/\overline{\alpha}_T\right)\right)} ,$$

which, together with (10), concludes the proof. $\qquad\square$

## D  Proof of the Lower Bound

In this section, we prove the lower bound provided in Section 5, which we restate below. As remarked before, our proof makes use of known techniques for proving lower bounds for the multitask bandit problem. In particular, parts of the proof are adapted from the proof of Theorem 7 in [17].

**Theorem 4.** *Pick any $K \geq 2$ and any $\alpha$ such that $2 \leq \alpha \leq K$. Then, for any algorithm and for all $T \geq \frac{\alpha \log_\alpha K}{4 \log(4/3)}$, there exists a sequence of losses and feedback graphs $G_1, \ldots, G_T$ such that $\alpha(G_t) = \alpha$ for all $t = 1, \ldots, T$ and*

$$R_T \geq \frac{1}{18\sqrt{2}}\sqrt{\alpha T \log_\alpha K}.$$

*Proof.* Once again, we define $M = \log_\alpha K$, which we assume for now to be an integer; we discuss in the end how to extend the proof to the case when it is not. The proof will be divided into five parts I–V. We begin by formalizing the class of environments described in Section 5 and stating two useful lemmas.

### I. Preliminaries

We remind the reader that we identify each action in $V$ with a vector $a = \left(a(1), \ldots, a(M)\right) \in [\alpha]^M$. We will focus on a set of $M$ undirected graphs $\mathcal{G} = \{G^i\}_{i=1}^M$, where $G^i$ consists of $\alpha$ isolated cliques (with self-loops) $\{C_{i,j}\}_{j=1}^\alpha$ such that an action $a$ belongs to clique $C_{i,j}$ if and only if $a(i) = j$. As remarked before, all these graphs have independence number $\alpha$. For convenience, we also use actions in $V$ as functions from $\mathcal{G}$ to $[\alpha]$, with $a(G^i) = a(i)$.

An environment is identified by a function $\mu\colon [\alpha] \times \mathcal{G} \to [0,1]$ such that at every round $t$, after having drawn a graph $G_t$ from the uniform distribution over $\mathcal{G}$ (denoted with $U_{\mathcal{G}}$), the environment latently draws for each $j \in [\alpha]$ and $G \in \mathcal{G}$, a Bernoulli random variable $\gamma_t(j; G)$ with mean $\mu(j; G_t)$. Subsequently, for defining the loss of action $a \in V$ at round $t$, we simply set $\ell_t(a) = \gamma_t(a(G_t); G_t)$, whose expectation, conditioned on $G_t$, is $\mu(a(G_t); G_t)$. To simplify the notation, we use $\mu(a; G)$ as shorthand for $\mu(a(G); G)$ and $\gamma_t(a; G)$ as shorthand for $\gamma_t(a(G); G)$. Denote by $A_t$ the action picked by the player at round $t$, which is chosen prior to observing $G_t$. We will focus on the following notion of stochastic regret, which we define for environment $\mu$ as:

$$\overline{R}_T(\mu) = \max_{a \in V} \mathbb{E}_\mu\left[\sum_{t=1}^{T}(\ell_t(A_t) - \ell_t(a))\right] \ ,$$

where $\mathbb{E}_\mu[\cdot]$ denotes the expectation with respect to the sequence of losses and graphs generated by environment $\mu$, as well as the randomness in the choices of the player. We can use the tower rule to rewrite this expression as

$$\overline{R}_T(\mu) = \max_{a \in V} \sum_{t=1}^{T} \mathbb{E}_\mu\left[\mathbb{E}_\mu\left[\mathbb{E}_\mu\left[\ell_t(A_t) - \ell_t(a) \,\Big|\, G_t, A_t\right] \Big| A_t\right]\right]$$

$$= \max_{a \in V} \sum_{t=1}^{T} \mathbb{E}_\mu\left[\mathbb{E}_\mu\left[\mu(A_t; G_t) - \mu(a; G_t) \,\Big|\, A_t\right]\right]$$

$$= \max_{a \in V} \sum_{t=1}^{T} \mathbb{E}_\mu\left[\sum_{i=1}^{M} U_{\mathcal{G}}(G^i)(\mu(A_t; G^i) - \mu(a; G^i))\right]$$

$$= \max_{a \in V} \frac{1}{M} \sum_{i=1}^{M} \mathbb{E}_\mu\left[\sum_{t=1}^{T}(\mu(A_t; G^i) - \mu(a; G^i))\right] \ . \tag{12}$$

For a fixed algorithm, one can show via standard arguments that

$$\sup_{(\ell_t)_{t=1}^{T}, (G_t)_{t=1}^{T}} R_T \geq \sup_\mu \overline{R}_T(\mu) \ .$$

Hence, it suffices for our purposes to prove a lower bound for the right-hand side of this inequality.

In the following, we will have to be more precise about the probability measure with respect to which the expectation in (12) is defined. Let $\boldsymbol{\lambda}_t \in \{0,1\}^{K/\alpha}$ denote the vector of losses observed by the player at round $t$, which corresponds to the losses of the actions connected to $A_t$ assuming that a systematic ordering of the actions makes it clear which coordinate of $\boldsymbol{\lambda}_t$ belongs to which action. Let $\mathbf{1}_{K/\alpha}$ and $\mathbf{0}_{K/\alpha}$ be the $K/\alpha$ dimensional[4] vectors of all ones and all zeros respectively. Clearly, we have that $\boldsymbol{\lambda}_t = \gamma_t(A_t; G_t)\mathbf{1}_{K/\alpha} = \ell_t(A_t)\mathbf{1}_{K/\alpha}$, which is a binary random variable taking values in $\{\mathbf{0}_{K/\alpha}, \mathbf{1}_{K/\alpha}\}$. Let $\mathbb{P}_\mu^\lambda$ be the probability distribution of $\boldsymbol{\lambda}_t$ in environment $\mu$. Notice then that we have that

$$\mathbb{P}_\mu^\lambda(\gamma_t = \mathbf{1}_{K/\alpha} \,|\, G_t = G, A_t = a) = \mu(a; G) \ . \tag{13}$$

Let $H_t = (A_1, G_1, \boldsymbol{\lambda}_1, \ldots, A_t, G_t, \boldsymbol{\lambda}_t) \in (V \times \mathcal{G} \times \{0,1\}^{K/\alpha})^t$ be the interaction trajectory after $t$ steps. The policy $\pi$ adopted by the player can be modelled as a sequence of probability kernels $\{\pi_t\}_{t=1}^{T}$ each mapping the trajectory so far to a distribution over the actions, i.e., $A_t$ is sampled from $\pi_t(\cdot \,|\, H_{t-1})$. An environment $\mu$ and a policy $\pi$ (implicit in the notation, and fixed throughout the rest of the proof) together define a distribution $\mathbb{P}_\mu$ over the set of possible trajectories of $T$ steps such that:

$$\mathbb{P}_\mu(H_T) = \prod_{t=1}^{T} \pi_t(A_t \,|\, H_{t-1}) U_{\mathcal{G}}(G_t) \mathbb{P}_\mu^\lambda(\lambda_t \,|\, G_t, A_t) \ .$$

If $P$ and $Q$ are two distributions defined on the same space, let $D_{\mathrm{KL}}(P \,\|\, Q)$ and $\delta(P, Q)$ be the KL-divergence and the total variation distance respectively between $P$ and $Q$. Furthermore, let $d(p \,\|\, q)$ be the KL-divergence between two Bernoulli random variables with means $p$ and $q$. The following lemma provides an expression for the KL-divergence between two the probability distributions associated to two environments.

---

[4]Note that $K/\alpha = \alpha^{M-1}$ is an integer since $M(\geq 1)$ was assumed to be an integer.

**Lemma 5.** *For a fixed policy, let $\mu$ and $\mu'$ be two environments as described above. Then,*

$$D_{\mathrm{KL}}(\mathbb{P}_\mu \,\|\, \mathbb{P}_{\mu'}) = \frac{1}{M} \sum_{i=1}^{M} \sum_{a \in V} N_\mu(a; T) d\big(\mu(a; G^i) \,\|\, \mu'(a; G^i)\big) \ ,$$

*where $N_\mu(a; T) = \mathbb{E}_\mu\Big[\sum_{t=1}^{T} \mathbb{I}\{A_t = a\}\Big]$.*

*Proof.* The proof is similar to that of Lemma 15.1 in [27]. Namely, we have in our case that

$$
\begin{aligned}
D_{\mathrm{KL}}(\mathbb{P}_\mu \,\|\, \mathbb{P}_{\mu'}) &= \mathbb{E}_\mu\left[\ln \frac{\mathbb{P}_\mu(H_T)}{\mathbb{P}_{\mu'}(H_T)}\right] \\
&= \mathbb{E}_\mu\left[\ln \frac{\prod_{t=1}^{T} \pi_t(A_t \mid H_{t-1}) U_{\mathcal{G}}(G_t) \mathbb{P}_\mu^\lambda(\lambda_t \mid G_t, A_t)}{\prod_{t=1}^{T} \pi_t(A_t \mid H_{t-1}) U_{\mathcal{G}}(G_t) \mathbb{P}_{\mu'}^\lambda(\lambda_t \mid G_t, A_t)}\right] \\
&= \sum_{t=1}^{T} \mathbb{E}_\mu\left[\ln \frac{\mathbb{P}_\mu^\lambda(\lambda_t \mid G_t, A_t)}{\mathbb{P}_{\mu'}^\lambda(\lambda_t \mid G_t, A_t)}\right] \\
&= \sum_{t=1}^{T} \mathbb{E}_\mu\left[\mathbb{E}_\mu\left[\mathbb{E}_\mu\left[\ln \frac{\mathbb{P}_\mu^\lambda(\lambda_t \mid G_t, A_t)}{\mathbb{P}_{\mu'}^\lambda(\lambda_t \mid G_t, A_t)}\,\Big|\, G_t, A_t\right]\,\Big|\, A_t\right]\right] \\
&= \sum_{t=1}^{T} \mathbb{E}_\mu\left[\mathbb{E}_\mu\left[D_{\mathrm{KL}}(\mathbb{P}_\mu^\lambda(\cdot \mid G_t, A_t) \,\|\, \mathbb{P}_{\mu'}^\lambda(\cdot \mid G_t, A_t))\,\Big|\, A_t\right]\right] \\
&= \sum_{t=1}^{T} \mathbb{E}_\mu\left[\sum_{i=1}^{M} U_{\mathcal{G}}(G^i) D_{\mathrm{KL}}(\mathbb{P}_\mu^\lambda(\cdot \mid G^i, A_t) \,\|\, \mathbb{P}_{\mu'}^\lambda(\cdot \mid G^i, A_t))\right] \\
&= \frac{1}{M} \sum_{i=1}^{M} \sum_{t=1}^{T} \mathbb{E}_\mu\left[D_{\mathrm{KL}}(\mathbb{P}_\mu^\lambda(\cdot \mid G^i, A_t) \,\|\, \mathbb{P}_{\mu'}^\lambda(\cdot \mid G^i, A_t))\right] \\
&= \frac{1}{M} \sum_{i=1}^{M} \sum_{a \in V} N_\mu(a; T) D_{\mathrm{KL}}(\mathbb{P}_\mu^\lambda(\cdot \mid G^i, a) \,\|\, \mathbb{P}_{\mu'}^\lambda(\cdot \mid G^i, a)) \\
&= \frac{1}{M} \sum_{i=1}^{M} \sum_{a \in V} N_\mu(a; T) d\big(\mu(a; G^i) \,\|\, \mu'(a; G^i)\big) \ ,
\end{aligned}
$$

where the last equality holds via (13). $\qquad\square$

The following standard lemma, adapted from [27], will be used in the sequel.

**Lemma 6.** *Let $P$ and $Q$ be probability measures on the same measurable space $(\Omega, \mathcal{F})$. Let $a < b$ and $X : \Omega \to [a, b]$ be an $\mathcal{F}$-measurable random variable. Then,*

$$\left|\int_\Omega X(\omega) dP(\omega) - \int_\Omega X(\omega) dQ(\omega)\right| \leq (b - a)\sqrt{\frac{1}{2} D_{\mathrm{KL}}(P \,\|\, Q)} \ .$$

*Proof.* We have, by Exercise 14.4 in [27], that

$$\left|\int_\Omega X(\omega) dP(\omega) - \int_\Omega X(\omega) dQ(\omega)\right| \leq (b - a)\delta(P, Q) \ ,$$

from which the lemma follows by applying Pinsker's inequality. $\qquad\square$

## II. Choosing the environments

We will construct a collection of environments $\{\mu_a\}_{a \in V}$, each associated to an action, such that for any $i \in [M]$ and $j \in [\alpha]$,

$$\mu_a(j; G^i) = \frac{1}{2} - \varepsilon \mathbb{I}\{a(i) = j\} \ ,$$

where $0 < \varepsilon \le \frac{1}{4}$ will be tuned later. In words, for a fixed graph, environment $\mu_a$ gives a slight advantage to actions that are connected to $a$ in that graph, and thus agree with $a$ in the corresponding game. Additionally, for every $a \in V$ and $i \in [M]$, we define environment $\mu_a^{-i}$ to be such that for any $s \in [M]$ and $j \in [\alpha]$,

$$\mu_a^{-i}(j; G^s) = \begin{cases} \frac{1}{2}, & \text{if } s = i \\ \mu_a(j; G^s), & \text{otherwise.} \end{cases}$$

Similar to [17], we will define, for every $i \in [M]$, an equivalence relation $\sim_i$ on the arms such that

$$a \sim_i a' \iff \forall s \in [M] \setminus \{i\}, a'(s) = a(s) \ ,$$

for any $a, a' \in V$. This means that two arms are equivalent according to $\sim_i$ if and only if their choices of base actions coincide in all games that are different from $i$. Let $V/\sim_i$ be the set of equivalence classes of $\sim_i$. It is easy to see that $V/\sim_i$ contains exactly $\alpha^{M-1}$ equivalence classes, and that each class consists of $\alpha$ actions, each corresponding to a different choice of base action in game $i$. Notice then that for an equivalence class $W \in V/\sim_i$, all environments $\mu_a^{-i}$ with $a \in V$ are indeed identical. In the sequel, this environment will also be referred to as $\mu_W^{-i}$.

## III. Lower-bounding the regret of a single environment

Note that in environment $\mu_a$, we have that $a = \arg\min_{a' \in V} \sum_{i=1}^M \mu_a(a'; G^i)$. Consequently, starting from (12) we get that

$$\begin{aligned}
\overline{R}_T(\mu_a) &= \sum_{i=1}^M \frac{1}{M} \mathbb{E}_{\mu_a}\left[\sum_{t=1}^T (\mu_a(A_t; G^i) - \mu_a(a; G^i))\right] \\
&= \sum_{i=1}^M \frac{1}{M} \mathbb{E}_{\mu_a}\left[\sum_{t=1}^T \left(\frac{1}{2} - \varepsilon\mathbb{I}\{A_t(i) = a(i)\} - \left(\frac{1}{2} - \varepsilon\right)\right)\right] \\
&= \frac{\varepsilon}{M} \sum_{i=1}^M \mathbb{E}_{\mu_a}\left[\sum_{t=1}^T (1 - \mathbb{I}\{A_t(i) = a(i)\})\right] \\
&= \frac{\varepsilon}{M} \sum_{i=1}^M \left(T - N_{\mu_a}(i, a; T)\right) \ ,
\end{aligned}$$

where for environment $\mu$, action $a$, and game $i$, $N_\mu(i, a; T) = \mathbb{E}_\mu\left[\sum_{t=1}^T \mathbb{I}\{A_t(i) = a(i)\}\right]$ is the expected number of times in environment $\mu$ that the action chosen by the policy agrees with action $a$ in game $i$. Next, we use Lemma 6 to obtain that

$$\overline{R}_T(\mu_a) \ge \frac{\varepsilon}{M} \sum_{i=1}^M \left(T - N_{\mu_a^{-i}}(i, a; T) - T\sqrt{\frac{1}{2}D_{\text{KL}}\left(\mathbb{P}_{\mu_a^{-i}} \,\|\, \mathbb{P}_{\mu_a}\right)}\right) \ . \tag{14}$$

For bounding the KL-divergence term, we start from Lemma 5:

$$
\begin{aligned}
D_{\mathrm{KL}}\big(\mathbb{P}_{\mu_a^{-i}} \,\big\|\, \mathbb{P}_{\mu_a}\big) &= \frac{1}{M}\sum_{s=1}^{M}\sum_{a'\in V} N_{\mu_a^{-i}}(a';T)d\big(\mu_a^{-i}(a';G^s) \,\big\|\, \mu_a(a';G^s)\big) \\
&= \frac{1}{M}\sum_{a'\in V} N_{\mu_a^{-i}}(a';T)d\big(\mu_a^{-i}(a';G^i) \,\big\|\, \mu_a(a';G^i)\big) \\
&= \frac{1}{M}\sum_{a'\in V} N_{\mu_a^{-i}}(a';T)d\big(1/2 \,\big\|\, 1/2 - \varepsilon\mathbb{I}\{a'(i) = a(i)\}\big) \\
&= \frac{1}{M}\sum_{a'\in V} \mathbb{I}\{a'(i) = a(i)\}N_{\mu_a^{-i}}(a';T)d\big(1/2 \,\big\|\, 1/2 - \varepsilon\big) \\
&\leq \frac{c\varepsilon^2}{M}\sum_{a'\in V} \mathbb{I}\{a'(i) = a(i)\}N_{\mu_a^{-i}}(a';T) \\
&= \frac{c\varepsilon^2}{M}\sum_{a'\in V} \mathbb{I}\{a'(i) = a(i)\}\,\mathbb{E}_{\mu_a^{-i}}\sum_{t=1}^{T}\mathbb{I}\{A_t = a'\} \\
&= \frac{c\varepsilon^2}{M}\,\mathbb{E}_{\mu_a^{-i}}\sum_{t=1}^{T}\mathbb{I}\{A_t(i) = a(i)\} \\
&= \frac{c\varepsilon^2}{M}N_{\mu_a^{-i}}(i,a;T) \ ,
\end{aligned}
$$

where the second equality holds since the two environments only differ in $G^i$, and the inequality holds for $\varepsilon \leq \frac{1}{4}$ with $c = 8\log\frac{4}{3}$. Plugging back into (14) gets us that

$$
\overline{R}_T(\mu_a) \geq \frac{\varepsilon}{M}\sum_{i=1}^{M}\left(T - N_{\mu_a^{-i}}(i,a;T) - \varepsilon T\sqrt{\frac{c}{2M}N_{\mu_a^{-i}}(i,a;T)}\right) \ . \tag{15}
$$

## IV. Summing up

Fix $i \in [M]$. Notice that for $W \in V/\sim_i$,

$$
\sum_{a\in W}\mathbb{I}\{A_t(i) = a(i)\} = 1
$$

since each action in $W$ corresponds to a different choice of base action in game $i$. Hence,

$$
\begin{aligned}
\frac{1}{\alpha^M}\sum_{a\in V} N_{\mu_a^{-i}}(i,a;T) &= \frac{1}{\alpha^M}\sum_{W\in V/\sim_i}\sum_{a\in W} N_{\mu_a^{-i}}(i,a;T) \\
&= \frac{1}{\alpha^M}\sum_{W\in V/\sim_i}\sum_{a\in W} N_{\mu_W^{-i}}(i,a;T) \\
&= \frac{1}{\alpha^M}\sum_{W\in V/\sim_i}\mathbb{E}_{\mu_W^{-i}}\left[\sum_{t=1}^{T}\sum_{a\in W}\mathbb{I}\{A_t(i) = a(i)\}\right] \\
&= \frac{1}{\alpha^M}\alpha^{M-1}T = \frac{T}{\alpha} \ .
\end{aligned}
$$

Using this together with (15) allows us to conclude that

$$
\sup_{\mu} \overline{R}_T(\mu) \geq \frac{1}{\alpha^M} \sum_{a \in V} \overline{R}_T(\mu_a)
$$

$$
\geq \frac{1}{\alpha^M} \sum_{a \in V} \frac{\varepsilon}{M} \sum_{i=1}^{M} \left( T - N_{\mu_a^{-i}}(i, a; T) - \varepsilon T \sqrt{\frac{c}{2M} N_{\mu_a^{-i}}(i, a; T)} \right)
$$

$$
\geq \frac{\varepsilon}{M} \sum_{i=1}^{M} \left( T - \frac{1}{\alpha^M} \sum_{a \in V} N_{\mu_a^{-i}}(i, a; T) - \varepsilon T \sqrt{\frac{c}{2M\alpha^M} \sum_{a \in V} N_{\mu_a^{-i}}(i, a; T)} \right)
$$

$$
= \frac{\varepsilon}{M} \sum_{i=1}^{M} \left( T - \frac{T}{\alpha} - \varepsilon T \sqrt{\frac{cT}{2M\alpha}} \right)
$$

$$
= \varepsilon T \left( 1 - \frac{1}{\alpha} - \varepsilon \sqrt{\frac{cT}{2M\alpha}} \right)
$$

$$
\geq \varepsilon T \left( \frac{1}{2} - \varepsilon \sqrt{\frac{cT}{2M\alpha}} \right) ,
$$

where the third inequality holds due to the concavity of the square root, and the last inequality holds by our assumption that $\alpha \geq 2$. Setting $\varepsilon = \frac{1}{4} \sqrt{\frac{2M\alpha}{cT}}$ yields that

$$
\sup_{\mu} \overline{R}_T(\mu) \geq \frac{1}{16} \sqrt{\frac{2}{c}} \cdot \sqrt{\alpha T M} \geq \frac{1}{18} \sqrt{\alpha T M} = \frac{1}{18} \sqrt{\alpha T \log_\alpha K} ,
$$

whereas it holds that $\varepsilon \leq \frac{1}{4}$ thanks to the assumption made on $T$ in the statement of the theorem.

## V. The case when $\log_\alpha K$ is not an integer

If $M$ is not an integer,[5] we can use the same construction as before for the first $\alpha^{\lfloor M \rfloor}$ actions and force the remaining actions to behave identically to some action in the construction. That is, we can designate a certain action such that, in all environments, all the excess actions receive the same loss as this action and are connected to it, to each other, and to every action that happens to share an edge with this designated action in a given graph (in other words, we are expanding the designated action into a clique). This way, the independence number of all the graphs in the construction is still $\alpha$, and the excess actions do not provide any extra utility to the learner; playing one of them is exactly like playing the designated action, and the construction does not hide this from the player. We can then obtain the same bound as before but in terms of $\lfloor M \rfloor$, thus costing us an extra $1/\sqrt{2}$ factor to recover the desired bound (using that $\lfloor M \rfloor \geq M/2$). $\qquad\square$

## E   Comparison with [14]

In [14], the authors consider a special case of the undirected feedback graph problem where the graph (fixed and known) is composed of $\alpha$ disjoint cliques with self-loops. For $j \in [\alpha]$, let $m_j$ denote the number of actions in the $j$-th clique, implying that $\sum_{j=1}^{\alpha} m_j = K$ (the number of arms). For this problem, [14, Theorem 4] provides a lower bound of order $\sqrt{T \sum_{j=1}^{\alpha} \ln(m_j + 1)}$. In particular, if the cliques are balanced (i.e., $m_1 = \cdots = m_\alpha = K/\alpha$), the lower bound becomes of order $\sqrt{\alpha T \ln(1 + K/\alpha)}$, thus matching the regret bound of Algorithm 1. This means that, for any value of $1 \leq \alpha \leq K$, there are feedback graphs on $K$ nodes with independence number $\alpha$ such that no other algorithm can achieve a better minimax regret guarantee than that of our proposed algorithm.

We emphasize that this does not imply *graph-specific* minimax optimality. Indeed, as shown in [14], when the cliques are unbalanced, the regret guarantee of our algorithm can be inferior to that of the algorithm they proposed, which matches the $\sqrt{T \sum_{j=1}^{\alpha} \ln(m_j + 1)}$ bound. However, beyond

---
[5]Note that $M$ is never smaller than 1 since $\alpha \leq K$.

the disjoint cliques case, their algorithm requires computing a minimum clique cover for the given feedback graph $G$, which is known to be NP-hard [24]. More importantly, their reliance on a clique cover leads to a dependence of the regret on the clique cover number $\theta(G)$ instead of the independence number $\alpha(G)$. One can argue that the ratio between $\theta(G)$ and $\alpha(G)$ can be $\Omega(K/(\ln K)^2)$ for most graphs on a sufficiently large number $K$ of vertices (e.g., see [29, Section 6]). Finally, it is not clear how to generalize their approach to time-varying feedback graphs (informed or uninformed). Hence, despite the contributions of our work and those of [14], the problem of characterizing the minimax regret rate at a graph-based granularity still calls for further investigation.

## F   Directed Strongly Observable Feedback Graphs

In this section, we consider the case of directed strongly observable graphs. For a directed graph $G = (V, E)$, let $N_G^{\text{in}}(i) = \{j \in V : (j, i) \in E\}$ be the in-neighbourhood of node $i \in V$ in $G$, and let $N_G^{\text{out}}(i) = \{j \in V : (i, j) \in E\}$ be its out-neighbourhood. A directed graph $G$ is strongly observable if for every $i \in V$, at least one of the following holds: $i \in N_G^{\text{in}}(i)$ or $j \in N_G^{\text{in}}(i)$ for all $j \neq i$. The independence number $\alpha(G)$ is still defined in the same manner as before; that is, the cardinality of the largest set of nodes such that no two nodes share an edge, regardless of orientation. The interaction protocol is the same as in the undirected case, except that, in each round $t \in [T]$, the learner only observes the losses of the actions in $N_{G_t}^{\text{out}}(I_t)$, which is the out-neighbourhood in graph $G_t$ of the action $I_t$ picked by the learner. As before, we will use $N_t^{\text{in}}(i)$ and $N_t^{\text{out}}(i)$ to denote $N_{G_t}^{\text{in}}(i)$ and $N_{G_t}^{\text{out}}(i)$ respectively. For this setting, a bound of $\mathcal{O}(\sqrt{\alpha T} \cdot \ln(KT))$ was proven in [2] for the EXP3.G algorithm. Later, [35] proved a bound of $\mathcal{O}(\sqrt{\alpha T (\ln K)^3})$ for OSMD with a variant of the $q$-Tsallis entropy regularizer where $q$ was chosen as $1 - 1/(\ln K)$.

To use Algorithm 1 in the directed case, one can define loss estimates analogous to (6) by using the in-neighbourhood in place of the neighbourhood in the relevant quantities. Namely, let $S_t = \{i \in V : i \notin N_t^{\text{in}}(i)\}$, $J_t = \{i \in S_t : p_t(i) > 1/2\}$, and $P_t(i) = \sum_{j \in N_t^{\text{in}}(i)} p_t(j)$. The loss estimates (again due to [35]) can then be given by

$$
\widehat{\ell}_t(i) = \begin{cases} \frac{\ell_t(i)}{P_t(i)} \mathbb{I}\left\{I_t \in N_t^{\text{in}}(i)\right\} & \text{if } i \in V \setminus J_t \\ \frac{\ell_t(i) - 1}{P_t(i)} \mathbb{I}\left\{I_t \in N_t^{\text{in}}(i)\right\} + 1 & \text{if } i \in J_t \end{cases} .
$$

Algorithm 1 with these loss estimates can be analyzed in a similar manner to the proof of Theorem 2, with the major difference being the way that the variance term is handled for actions with self-loops. Namely, the relevant term is

$$
\sum_{i \in V : i \in N_t^{\text{in}}(i)} \frac{p_t(i)^{2-q}}{\sum_{j \in N_t^{\text{in}}(i)} p_t(j)},
$$

on which we elaborate more in the following.

Let $p \in \Delta_K$ and $\beta \in (0, 1/2)$ be such that $\min_{i \in V} p(i) \geq \beta$. We first consider the variance term given by the negative Shannon entropy regularizer. It is known [2] that such a variance term, restricted to nodes with a self-loop in the strongly observable feedback graph $G = (V, E)$, has an upper bound of the form

$$
\sum_{i \in V : i \in N_G^{\text{in}}(i)} \frac{p(i)}{\sum_{j \in N_G^{\text{in}}(i)} p(j)} \leq 4\alpha(G) \ln\left(\frac{4K}{\alpha(G)\beta}\right) . \tag{16}
$$

In addition to the fact that this variance bound has a linear dependence on the independence number $\alpha(G)$ of $G$, we observe that there is a logarithmic factor in $K/\alpha$ and $1/\beta$ given by the fact that we now consider directed graphs. The main problem is that, in general, we cannot hope to improve upon the above logarithmic factor as it can be shown to be unavoidable unless we manage to restrict the probability distributions we consider. Indeed, it is possible to show [3, Fact 4] that there exist probability distributions $p \in \Delta_K$ and directed strongly observable graphs $G$ for which $\alpha(G) = 1$ and

$$
\sum_{i \in V : i \in N_G^{\text{in}}(i)} \frac{p(i)}{\sum_{j \in N_G^{\text{in}}(i)} p(j)} = \frac{K+1}{2} = \frac{1}{2} \log_2\left(\frac{4}{\min_i p(i)}\right) = \alpha(G) \log^{\omega(1)}\left(\frac{K}{\alpha(G)}\right) .
$$

A usual way to avoid this is to introduce some explicit exploration to the probability distributions in order to force a lower bound on the probabilities of all nodes, e.g., as in EXP3.G [2]. This would bring

the linear dependence on $K$ down to $\alpha$ in the above bad case, while, on the other hand, introducing a $\ln(KT)$ factor which then worsens the overall dependence on the time horizon $T$.

Consider now the variance term given by the analysis of the $q$-FTRL algorithm. As already argued in Section 3, we can reuse the variance bound in (16) for the case of negative Shannon entropy because

$$\sum_{i \in V : i \in N_G^{\text{in}}(i)} \frac{p(i)^{2-q}}{\sum_{j \in N_G^{\text{in}}(i)} p(j)} \leq \sum_{i \in V : i \in N_G^{\text{in}}(i)} \frac{p(i)}{\sum_{j \in N_G^{\text{in}}(i)} p(j)}$$

for any $q \in (0, 1)$, and such a bound is the best known so far for the general case of directed strongly observable graphs. However, we can be more clever in the way we utilize it. Similarly to the proof of [35, Theorem 14], we can gain an advantage from the adoption of $q$-FTRL by splitting the sum in the variance term into two sums according to some adequately chosen threshold $\beta$ on the probabilities of the individual nodes. More precisely, by choosing $\beta \approx \exp\big(-\ln(K/\alpha)\ln K\big)$ and $q = 1 - 1/(\ln K)$, we can prove that

$$\sum_{i \in V : i \in N_G^{\text{in}}(i)} \frac{p(i)^{2-q}}{\sum_{j \in N_G^{\text{in}}(i)} p(j)} = \mathcal{O}\left(\alpha \ln K \left(1 + \ln \frac{K}{\alpha}\right)\right) \ .$$

We can further argue that, by following a similar analysis as in the proofs of Theorems 1 and 2, this variance bound would allow to show that the regret of $q$-FTRL is $\mathcal{O}\left(\sqrt{\alpha T \left(1 + \ln(K/\alpha)\right)} \cdot \ln K\right)$, where there is an additional $\ln K$ factor when compared to our regret bound in the undirected case (Theorem 2).

The presence of extra logarithmic factors is to be expected in the directed case, as many edges between distinct nodes might reduce the independence number of the graph, while providing information in one direction only. However, the undirected graph $G'$ obtained from any directed strongly observable graph $G$ by reciprocating edges between distinct nodes has the same independence number $\alpha(G') = \alpha(G)$ but the regret guarantee given by the more general analysis of $q$-FTRL would introduce a spurious $\ln K$ multiplicative factor. We remark that all the currently available upper bounds on the variance term (either with negative Shannon entropy or negative $q$-Tsallis entropy regularizers) do not exactly reflect the phenomenon of a gradually disappearing logarithmic factor when the graph is closer to being undirected (i.e., has fewer unreciprocated edges).

Taking these observations into account, we believe that it should be possible to achieve tighter guarantees that match our intuition, by improving the currently available tools. The bound on the variance term, for instance, is one part of the analysis that might be improvable. We might want to have a similar bound as (16) but with a sublinear dependence on $\alpha$ that varies according to the parameter $q$ of the negative $q$-Tsallis entropy; e.g., ignoring logarithmic factors, we could expect it to become of order $\alpha^q$ as we managed to prove for the undirected case (Lemma 1). Doing so could allow a better tuning of $q$ that might lead to improved logarithmic factors in the regret.