# OpenReview forum: "On the Minimax Regret for Online Learning with Feedback Graphs"
_NeurIPS.cc/2023/Conference — NeurIPS 2023 spotlight_

### Official Review · Reviewer_J2zm · 2023-06-15

**Soundness:** 4 excellent
**Presentation:** 4 excellent
**Contribution:** 3 good
**Rating:** 7
**Confidence:** 4

**Summary:**

This paper considers a classic problem of online learning with feedback graphs, which interpolates the full-information feedback and the bandit feedback. Specifically, the authors consider the case where the feedback graph is undirected, meaning that if node $i$ can observe node $j$, then node $j$ can observe node $i$. The best known $T$-dependent upper bound for this problem is $O(\sqrt{\alpha T\ln K})$ by [Alon et al., 2013]. For general strongly-observable graph, the best known $T$-dependent upper bound is $O(\sqrt{\alpha T\ln^3 K})$ proposed in [Zimmert and Lattimore, 2019]. In this paper, the authors propose an FTRL algorithm with q-Tsallis entropy regularization, showing $O(\sqrt{\alpha T(1+\ln \frac{K}{\alpha})})$ upper bound. This recovers the minimax regret bound in both bandit $O(\sqrt{KT})$ regret and full-information setup $O(\sqrt{T\ln K})$. Moreover, the authors also propose an improved $\Omega(\sqrt{\alpha T\frac{\ln K}{\ln \alpha}})$ lower bound compared with the $\Omega(\sqrt{\alpha T})$ lower bound proven in [Alon et al., 2015], though this lower bound does not match the upper bound obtained by the q-Tsallis FTRL algorithm. In addition, the authors also generalize their results to the undirect strongly observable graphs and time-varying graphs.

**Strengths:**

- The paper is well-written and the proposed algorithm is easy to follow.
- The designed algorithm improves upon the best known $T$-dependent regret bound for undirected strongly observable graphs. Specifically, the important technical contribution is Lemma 1, which shows that the stability term of the FTRL can be bounded by $\alpha^{1+b}$ where $b$ is decided by the parameter choice of Tsallis-entropy, compared to the one with log factors obtained in [Alon et al., 2015] and [Zimmert and Lattimore, 2019].
- The lower bound also tries to bridge the previous gap between the full-information case and the feedback graph case, although there is still a gap between the upper and the lower bound and the technical used in the lower bound is very similar to the one used for proving the lower bound for contextual bandit in [Seldin and Lugosi, 2016].

**Weaknesses:**

I do not find major weaknesses of this paper, except that the current upper bound is only obtained for the undirected strongly observable graph, which is also discussed by the authors in the appendix. Generalizing this technique to the general strongly observable graphs would be interesting.

**Questions:**

- The current algorithm is an FTRL-based algorithm. I wonder whether a certain type of online mirror descent based algorithm with a time-varying learning rate can also achieve similar results?

**Limitations:**

See Weakness and Questions.

---

> ### Author Rebuttal · Authors · 2023-08-07
>
> We thank the reviewer for the feedback. As mentioned in our answer to Reviewer TWzC, we hope the case of directed feedback graphs could be addressed by extending the proposed techniques. Regarding your question, OMD could be used in place of FTRL with the same techniques presented in this work. With regards to the adoption of a time-varying learning rate, we remark that adapting to a sequence of graphs with arbitrary independence numbers would also require the Tsallis entropy parameter to adapt to such a sequence, which poses significant challenges for the analysis. This is why we adopted an approach based on the doubling trick.

---

> > ### Comment · Reviewer_J2zm · 2023-08-11
> > **Thanks**
> >
> > Thanks for the authors' response. The response addresses my questions and I keep the original score.

---

### Official Review · Reviewer_TWzC · 2023-07-04

**Soundness:** 3 good
**Presentation:** 3 good
**Contribution:** 4 excellent
**Rating:** 8
**Confidence:** 2

**Summary:**

The paper investigates no-regret online learning algorithms performing under the feedback graph model. More precisely, at each round $t$ a learner selects an action (out of set of possible actions) incurring the cost associated with the specific action at the specific round. The actions are additionally vertices of an undirected graph that changes from round to round. Once the action $i$ is selected at round $t$, the learner is also informed on the costs of the neighboring actions at the specific rounds. The latter setting generalizes the bandit setting (the respective graph is composed by isolated vertices) as well as the full-information setting (the respective graph is a clique). Initially the authors consider the special case where every graph at each round admits independence number $\alpha$ ($\alpha =1$ for cliques and $\alpha =K$ for graphs with $K$ isolated edges). The authors first provide a nearly tight regret guarantee ($O(\sqrt{\alpha T \log (k/\alpha)})$) for the above special case. The authors also provide a nearly matching $\Omega(\sqrt{\alpha T \log K / \log \alpha})$ lower bound on the regret. Finally the authors remove the assumption that all the arriving graphs admit the same independence number and provide an $O(\sqrt{\overline{\alpha} T \log (k/\overline{\alpha})})$ regret bound where $\overline{a}$ is the time-average independence number of the graphs.

**Strengths:**

I think the paper is very interesting both from the perspective of results as well as from the perspective of techniques. I find very interesting the fact that the provided regret bound matches the respective regret bounds for the bandit ($\alpha = K$) and the full-information case ($\alpha=1$).  Also the authors provide a nearly matching lower bound that is based on a novel reduction to multitask learning. Both the techniques for the upper and lower bound require no trivial ideas that are nicely illustrated in the current write up. I also liked the fact that the provided algorithm can be modified to handle the case of time-varying independence numbers.

I overall believe that the paper provides very solid results and that will be of great interest of the online learning audience of NeurIPS.

**Weaknesses:**

The only limitation of the paper that I can find is the assumption on undirected graphs. The latter is also signified by the authors and considering the complexity of the current I think it is more than fair to consider the directed case for future work.

**Questions:**

-

---

> ### Author Rebuttal · Authors · 2023-08-07
>
> We thank the reviewer for the feedback. As mentioned in our answer to Reviewer J2zm, we agree that the directed feedback graphs case is an interesting future direction, which we hope could be addressed by extending the proposed techniques.

---

> > ### Comment · Reviewer_TWzC · 2023-08-11
> >
> > Thank you for your response. After reading the other reviews, I am confident that this is a very good paper and I have decided to keep my current score.

---

### Official Review · Reviewer_GSpg · 2023-07-05

**Soundness:** 4 excellent
**Presentation:** 4 excellent
**Contribution:** 2 fair
**Rating:** 7
**Confidence:** 4

**Summary:**

Two classical learning problems are learning with experts and multi-armed bandits. In both problems, a learner interacts during $T$ rounds with a set of $K$ actions by selecting an action at each round. After each round, the loss/reward at that step is revealed for all actions (resp. only for the selected action) in the expert (resp. bandit) problem. Mannor and Shamir [24] provided a convenient interpolation between the two problems by considering a generalization via feedback graphs. At each step $t$, losses are observed only at neighbor actions to the selected one, according to a graph $G_t$. The expert case corresponds to the clique, while the bandit case corresponds to graphs without edges. In this context, it is known that if the graphs have independence number $\alpha$, the minimax regret lies between $O(\sqrt{\alpha T\log K})$ and $\Omega(\sqrt{\alpha T})$. Further, the lower (resp. upper) bound is tight for learning with experts (resp. multi-armed bandits) for which $O(\sqrt{KT})$ regret is achievable (resp. a lower bound $\Omega(\sqrt{KT\log K})$ is known).


This paper investigates the corresponding $\sqrt{\log K}$ gap between the case $\alpha=1$ (clique/experts) and $\alpha=K$ (multi-armed bandits) and gives interpolating upper bound $O(\sqrt{\alpha T(1+\log K/\alpha)})$ on the regret provided that the graphs are undirected and strongly observable (each vertex has either a self-loop or neighbor to all other actions), with the same algorithm---FTRL with $q$-Tsallis entropy---run with appropriate parameter $q(\alpha)$ and loss estimates. Via a doubling trick, they show that one can achieve this bound up to an additive $\log \alpha$ term without prior knowledge on $\alpha$ (and can be further extended to the case when graphs $G_t$ do not share the same independence number).

Last, the authors provide an improved interpolating lower bound $\Omega(\sqrt{\alpha T \log K/\log \alpha})$, which requires the graphs $G_t$ to vary over time (but with same independence number), showing that when $\alpha$ is sub-polynomial in $K$, learning with feedback graphs requires an extra factor compared to the known lower bound $\Omega(\sqrt{\alpha T})$. This is shown via a reduction to the multitask bandit problem (different tasks are simulated by changing the graph over time).

**Strengths:**

The paper is very well-written and pleasant to read. The question is well posed, and motivated by the fact that improving the $\sqrt{\log K}$ factor for learning with experts, from $O(\sqrt{KT\log K})$ to $O(\sqrt{KT})$ remained open for a significant amount of time in the literature. Hence, interpolating the regret bounds in terms of the remaining factor $\sqrt{\log K}$ between the known bounds $O(\sqrt{\alpha T\log k})$ and $\Omega(\sqrt{\alpha T})$ seems an important question.

The upper bounds use the same algorithm (FTRL with $q$-Tsallis entropy) with adapted parameters $q$, which also interpolates between known approaches for the extreme cases $\alpha=1$ or $\alpha=K$, giving new insights on the impact of this parameter on the learning behavior.

**Weaknesses:**

Although the paper presents new contributions to interpolate between experts and bandits, the proofs follow classical arguments in the literature and seem to bring limited original ideas. In particular, it was known that the $q$-Tsallis entropy specifically allows achieving the minimax entropy in both end cases. The main difficulty in the generalization to feedback graphs is in a novel bound for the variance term in the FTRL analysis (Lemma 3), which itself is a generalization of known results in the literature for $q=1$. The rest of the proof follows standard analysis.

The lower bound heavily relies on the fact that graphs are allowed to vary over time. It seems that the approach cannot be extended to the important case of interest of a fixed graph (which seemed to be the main case studied in the previous literature on feedback graph bandits). The paper would also be significantly strengthened if the lower bound could be improved to match the upper bound when e.g. the independence number $\alpha$ grows as a polynomial in $K$---in that case, the present paper does not provide improvements in terms of rates compared to the existing literature. As such, the gap left by this paper seems to remain quite important.

**Questions:**

- Does the analysis carry over for adaptive adversaries, or is the oblivious assumption important?
- l.97 the independence number is the cardinality of such a set, right?
- What happens if we assume that the graph is constant over time? Would you expect similar lower bounds to hold?

**Limitations:**

The limitations were very well addressed by the authors, who clearly identify the new bound on the variance (Lemma 3) as the key added technical contribution for the upper bounds.

---

> ### Author Rebuttal · Authors · 2023-08-07
>
> We thank the reviewer for their feedback. We remark that obtaining the key result of Lemma 1 was an obstacle for prior work in the way of obtaining improved guarantees using $q$-Tsallis entropy with feedback graphs. Although we focused on the case of time-varying graphs in our lower bound construction, we conjecture that the upper bound is tight for all values of $K$ and $\\alpha$, even in the case of some fixed graph. In any case, please note that the lower bound we provided is the first such result that hints at the necessity of a logarithmic factor for the minimax regret of this problem (beyond the experts case). To address the question about adaptive adversaries, the analysis does carry over straightforwardly if the adversary adapts to the previous choices of the learner with our current notion of regret, but it would be interesting to investigate the more challenging notions of dynamic or adaptive regret. Finally, we thank the reviewer for pointing out the typo.

---

> > ### Comment · Reviewer_GSpg · 2023-08-14
> >
> > Thank you for your answers and comments. After re-examing the paper, I agree that the contributions are significant and the new bounds in the lemmas are significant. I have updated my score as a result.

---

### Official Review · Reviewer_kfEi · 2023-07-08

**Soundness:** 3 good
**Presentation:** 3 good
**Contribution:** 3 good
**Rating:** 7
**Confidence:** 4

**Summary:**

The paper studies the online learning under partial observations given by an underlying graph structure. This model is a common generalization of the bandit model (where the feed graph just contains self loops) and the full information model (where the graph is a complete graph with self loops). This model has been considered in the literature where it has been shown that the regret (under a technical condition known as strong observability) is $\tilde{\Theta} ( \sqrt{ \alpha T } )$ where $T$ is the time horizon and $\alpha$ is the independence number of the graph. Prior to the present paper there was a logarithmic gap in the regret upper and lower bounds. In particular, the previous bound when instantiated in the case of bandits led to an extra $log K$ factor. The present paper addresses this gap by designing an algorithm  that achieves regret $\sqrt{ \alpha T (1 + \log (K/\alpha)) }$ which gives the optimal bounds in both extremes. Furthermore, the paper presents lower bounds matching their given bounds.

The algorithm that the paper analyses is a version of FTRL with the Tsallis entropy regularizer. Since the choice of $q=1/2$ is known to be optimal for bandits and $q=1$ is optimal for full information, the natural idea is to interpolate between using a $\alpha$ dependent choice of $q$. The paper analyses the variance of the natural inverse probability weighted estimator of the loss in a novel way to arrive at the optimal choice to $q$ to get the regret bound.

The paper extends their analysis to a slightly more general settings (than just graphs with self loops; time varying independence numbers).

**Strengths:**

As stated in the summary, the main technical novelty is the analysis of the variance of the IPW estimator in the setting of feedback graphs and FTRL. This is itself a rather interesting contribution and I expect this to be useful in various other contexts. The main result presented in the paper resolves an interesting and natural question in the literature.

**Weaknesses:**

One main drawback of the approach of $\alpha$ dependent choice of the $q$ is that it makes the algorithm extremely inefficient (in theory at least). In particular, the independence number of a graph is known to (extremely) hard to approximate; in particular, unless $P=NP$ there is no "non-trivial" approximation algorithm even. This is to be contrasted with the previous work in the feedback graph setting which are efficient ( polynomial in $N$ the size of the graph/number of bandits). It remains an excellent open question to see if there is an efficient algorithm achieving this bound.

**Questions:**

- Is it immediately clear how the regret behaves when one only has an estimate of the independence number instead of the exact quantity?

**Limitations:**

Yes

---

> ### Author Rebuttal · Authors · 2023-08-07
>
> We thank the reviewer for their feedback. It is indeed true that computing, or even approximating, $\\alpha$ is computationally intractable in general. Please note that we do address this issue in Section 4, where we not only generalize our approach to time-varying feedback graphs with possibly different independence numbers, but also lift the requirement of knowing these independence numbers. If we specialize this setting to that of a fixed graph, we can avoid requiring knowledge of $\\alpha$ altogether. We thank the reviewer for pointing out this possible misunderstanding. We presented the first result assuming the knowledge of $\\alpha$ to simplify the presentation, but we will make sure to clarify this important feature of our theory in the revised version.

---

> > ### Comment · Reviewer_kfEi · 2023-08-18
> >
> > We thank the authors for pointing this out and apologize for the oversight. Increasing the score accordingly.

---

### Decision · Program_Chairs · 2023-09-21

**Decision:**

Accept (spotlight)

**Comment:**

There is unanimous support for this paper. An important innovation is a technically novel analysis of a key step from the now-standard Tsallis entropy-based regret analysis (specifically, bounding a certain variance-like quantity in such a way as to get dependence on the independence number rather than the number of actions). There initially were concerns about the level of technical innovation, but the from the discussion period with the authors, the concerned reviewers agreed about the technical novelty. While ultimately, this work "only" amounts to refining bounds at the level of log factors, such refinements often are not easy, which makes this contribution impressive. This work will be a welcome contribution to this year's conference.